# Th17 cells contribute to combination MEK inhibitor and anti-PD-L1 therapy resistance in *KRAS/p53* mutant lung cancers

David H. Peng [1,2,8], B. Leticia Rodriguez[1,8], Lixia Diao [3], Pierre-Olivier Gaudreau[1,4], Aparna Padhye [1,5], Jessica M. Konen[1], Joshua K. Ochieng [1], Caleb A. Class[6], Jared J. Fradette [1], Laura Gibson [1], Limo Chen[1], Jing Wang [3], Lauren A. Byers [1] & Don. L. Gibbons [1,7✉]

Understanding resistance mechanisms to targeted therapies and immune checkpoint blockade in mutant KRAS lung cancers is critical to developing novel combination therapies and improving patient survival. Here, we show that MEK inhibition enhanced PD-L1 expression while PD-L1 blockade upregulated MAPK signaling in mutant KRAS lung tumors. Combined MEK inhibition with anti-PD-L1 synergistically reduced lung tumor growth and metastasis, but tumors eventually developed resistance to sustained combinatorial therapy. Multi-platform profiling revealed that resistant lung tumors have increased infiltration of Th17 cells, which secrete IL-17 and IL-22 cytokines to promote lung cancer cell invasiveness and MEK inhibitor resistance. Antibody depletion of IL-17A in combination with MEK inhibition and PD-L1 blockade markedly reduced therapy-resistance in vivo. Clinically, increased expression of Th17-associated genes in patients treated with PD-1 blockade predicted poorer overall survival and response in melanoma and predicated poorer response to anti-PD1 in NSCLC patients. Here we show a triple combinatorial therapeutic strategy to overcome resistance to combined MEK inhibitor and PD-L1 blockade.

[1] Department of Thoracic/Head and Neck Medical Oncology, The University of Texas MD Anderson Cancer Center, Houston, TX, USA. [2] Perlmutter Cancer Center, NYU Langone Health, 550 First Avenue, Smilow Building 10th Floor, Suite 1010, New York, NY, USA. [3] Department of Bioinformatics and Computational Biology, The University of Texas MD Anderson Cancer Center, Houston, TX, USA. [4] Thoracic & Upper GI Cancer Research Laboratories, Research Institute of the McGill University Health Centre, Montreal, QC, Canada. [5] The University of Texas MD Anderson Cancer Center UT Health Graduate School of Biomedical Sciences, Houston, TX, USA. [6] Department of Biostatistics, The University of Texas MD Anderson Cancer Center, Houston, TX, USA. [7] Department of Molecular and Cellular Oncology, The University of Texas MD Anderson Cancer Center, Houston, TX, USA. [8] These authors contributed equally: David H. Peng, B. Leticia Rodriguez. ✉email: dlgibbon@mdanderson.org

Non-small cell lung cancer (NSCLC) is the leading cause of cancer-related deaths owing to late-stage disease presentation, metastasis, and resistance to conventional therapies. Approximately 30% of patients with lung adenocarcinoma possess an activating KRAS mutation, which currently lacks approved pharmacological drugs to effectively target all variants of the oncogenic protein[1,2]. Although MEK is a canonical downstream effector of activated mutant KRAS in the MAPK signaling pathway, MEK inhibitors have failed to yield clinical benefit in KRAS mutant cancers[3,4]. Previous studies have demonstrated that the acquisition of common secondary mutations, such as p53, promotes resistance to MEK inhibitors[5]. Additional studies by our group utilizing *Kras;p53* (KP) mutant mouse lung tumor models[6] demonstrate that epithelial subpopulations of lung cancer cells are responsive to MEK inhibitors, whereas drug-resistant lung cancer cells undergo a ZEB1-dependent epithelial-to-mesenchymal transition (EMT)[7,8]. Conversely, our prior studies also demonstrate that mesenchymal KP lung tumors are more responsive to PD-L1/PD-1 axis immune checkpoint blockade compared with epithelial KP tumors, owing to a ZEB1-mediated upregulation of PD-L1 and other checkpoint proteins in mesenchymal cells[9–11].

Although the implementation of PD-1 or PD-L1 immune checkpoint blockade has significantly improved lung cancer patient survival, only a minority of patients show durable response to treatment, suggesting innate or acquired resistance to immunotherapies[12]. Our reported findings suggest that the two distinct subpopulations of lung cancer cells have complementary responses to the individual treatments, providing a potential rationale to combine MEK inhibition with immune checkpoint blockade to overcome resistance to the individual therapies, complementing an on-going clinical trial at MD Anderson (ClinicalTrials.gov Identifier: NCT03225664). Previous clinical trials combining MEK inhibitor with anti-PD-L1 in solid tumors (melanoma, NSCLC, and colorectal cancers) show a manageable safety profile, but with only moderate tumor response[13–15]. Taken together, these trials have demonstrated disappointing results, even in the KRAS and BRAF mutant subgroups, and despite a demonstrated increase in CD8+ T-cell infiltration into tumors with the treatment, suggesting that other secondary factors may limit the efficacy of the dual treatment. Thus, performing murine pre-clinical trials with MEK inhibitors and PD-L1 blockade will elucidate potential resistance mechanisms and identify additional therapeutic targets.

Here, we first show that the combination of MEK inhibition with PD-L1 blockade significantly reduced KP lung tumor growth and metastasis compared with monotherapy treatments. We observed that the initial response to the drug combination was unsustainable with long-term treatment, as primary lung tumors eventually developed resistance. Cytokine array profiling revealed that resistant tumors had increased infiltration of Th17 CD4+ T cells, which secrete the tumor-promoting cytokines IL-17 and IL-22[16]. Antibody depletion of IL-17A in combination with MEK inhibition and PD-L1 blockade produced a durable reduction in lung tumor growth, metastasis, and prevented the development of tumor resistance. Gene expression analysis of melanoma patients and NSCLC patients treated with PD-1 blockade revealed that increasing levels of Th17-associated gene signatures predicted poorer overall survival and response to immune checkpoint blockade. Our findings reveal the molecular rationale for combining MEK inhibitors with PD-L1 blockade, identify the mechanism of combinatorial drug resistance, identify potential predictive markers of immunotherapy response, and validate a promising triple combinatorial treatment strategy for patients with KRAS mutant lung cancer.

## Results

**MEK inhibition increases PD-L1 expression while PD-L1 blockade upregulates MAPK signaling.** Previous work from our laboratory demonstrated that epithelial subpopulations of mutant KRAS lung cancers are responsive to MEK inhibitors while mesenchymal cells within the tumors are resistant[7,8]. Therefore, we sought to identify potential molecular targets that are specific to mesenchymal subpopulations to synergize with MEK inhibitor treatment. We utilized reverse phase protein array (RPPA) analysis[17,18] of heterogeneous syngeneic 344SQ KP lung tumors previously treated with the MEK inhibitor selumetinib (AZD6244)[7] to identify differentially regulated signaling proteins following MEK inhibition. RPPA profiling revealed a significant (false discovery rate (FDR) < 0.05) upregulation of CD274 (PD-L1) in the 344SQ KP tumors when tumor-bearing mice were treated with AZD6244 (Fig. 1a). Validation of the RPPA data by western blotting confirmed upregulation of PD-L1 (Fig. 1b). To further test our observation, we analyzed MEK inhibitor-sensitive 393P KP tumors that were previously treated with AZD6244 and harvested at either the point of drug sensitivity (393P AZD-S) or after the development of resistance (393P AZD-R)[7]. Consistent with our prior findings[7,19], MEK inhibitor-resistant 393P tumors showed an upregulation of the EMT-associated transcription factor ZEB1 along with increased PD-L1 expression (Fig. 1c). Interestingly, 393P tumors showed an increase in PD-L1 expression even at the point of sensitivity to MEK inhibition (Fig. 1c). To determine whether the increase in PD-L1 expression is tumor cell-intrinsic rather than a secondary signal from other cell types present in tumor tissues, we treated murine KP (393P and 344SQ) and human (H2122, H1299, H358, H157, and A549) lung cancer cell lines harboring varying co-mutations in vitro with AZD6244 and observed an increase in PD-L1 mRNA and protein expression in lung cancer cells following MEK inhibition (Fig. 1d, e; Supplementary Fig. 1a, b). Interestingly, cell lines (H157 and A549) harboring an LKB1 mutation (KL) had a much lower overall expression of PD-L1 as compared with cell lines (H2122, H1299, and H358) harboring an Kras/p53 mutation (Supplementary Fig. 1a).

Previous work by our group demonstrated that mesenchymal KP tumors express high levels of PD-L1 and are responsive to PD-L1 blockade during early phases of tumor growth before developing resistance to immunotherapy[19,20]. Similarly, to identify upregulated protein targets following PD-L1 blockade resistance, we analyzed RPPA data from 344SQ tumors that previously developed resistance to anti-PD-L1 blockade antibody after 7 weeks of treatment[20]. We observed an increase in the signaling of multiple pathways, including activated phosphorylated MAPK signaling proteins (Fig. 1f), which was validated by western blotting (Fig. 1g). To determine whether the increase in MAPK signaling following PD-L1 blockade was owing to an immune-mediated response, we performed a co-culture assay with splenocytes and observed an increase in tumor cell MAPK signaling only when 344SQ cells were co-cultured with splenocytes and treated with anti-PD-L1 (Supplementary Fig. 1c). Our findings demonstrate a reciprocal activation of PD-L1 and MAPK signaling when tumors are treated with single-agent MEK inhibitor or PD-L1 blockade, respectively.

**Combination MEK inhibition and PD-L1 blockade initially reduces lung tumor growth and metastasis, but ultimately therapeutic resistance develops.** As MEK inhibition increases PD-L1 expression while PD-L1 blockade enhances MAPK signaling in lung tumors, we next sought to combine the two treatments to prevent outgrowth of resistant subpopulations from the individual therapies. The combination of AZD6244

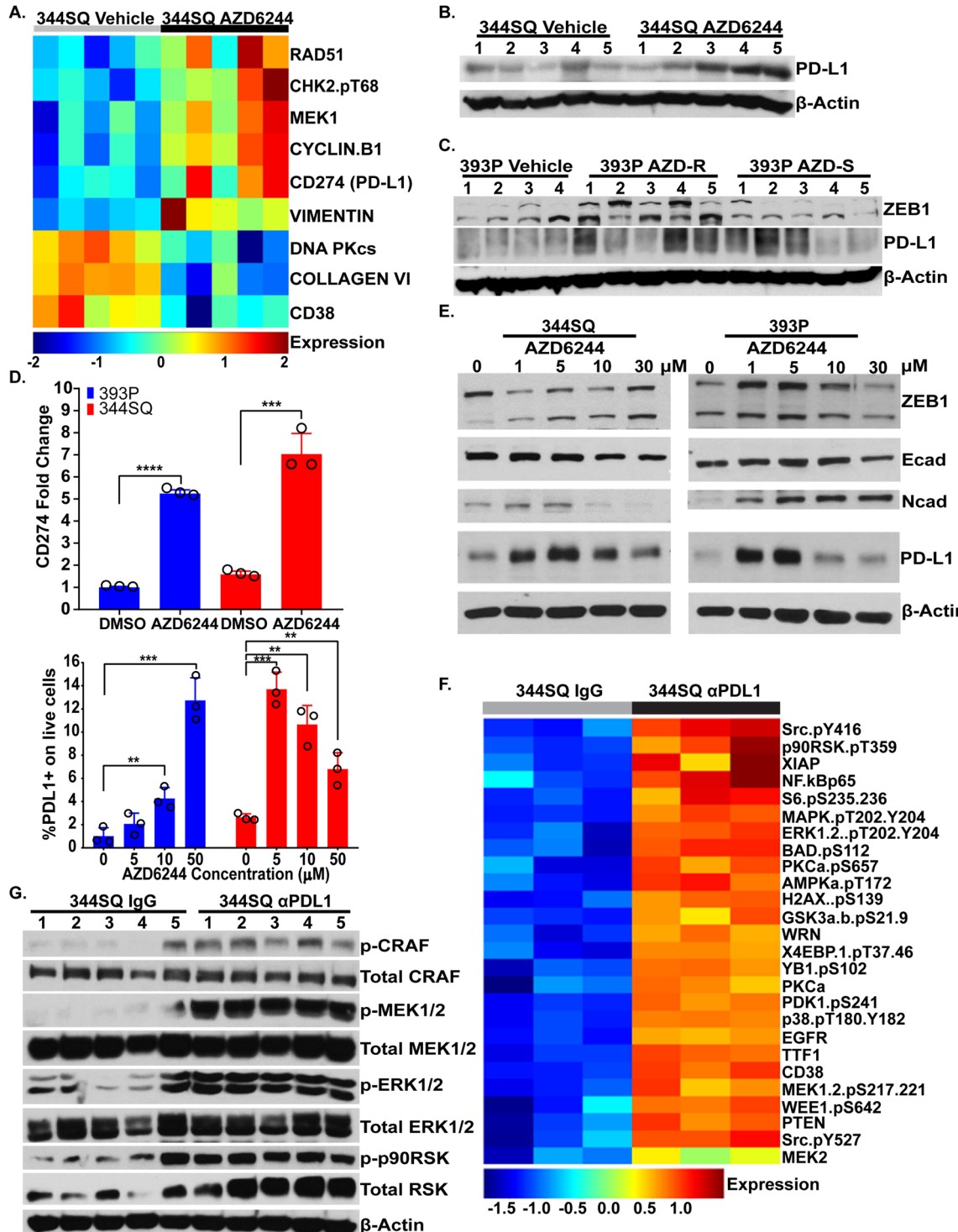

**Fig. 1 MEK inhibition increases PD-L1 expression while PD-L1 blockade upregulates MAPK signaling. A** Heatmap of reverse phase protein array (RPPA) profile showing statistically significant (FDR < 0.05) differentially regulated proteins in 344SQ murine KP syngeneic tumors treated with selumetinib (AZD624) MEK inhibitor (black) or vehicle (grey) for 4 weeks. **B** Western blot of PD-L1 and β-actin in 344SQ tumors treated with vehicle or AZD6244 for 4 weeks. Number indicates tumor replicate. **C** Western blot of ZEB1, PD-L1, and β-actin in 393P tumors treated with AZD6244 for 4 weeks or 8 weeks when tumors were sensitive (AZD-S) or resistant (AZD-R) to MEK inhibition, respectively. Number indicates tumor replicate. **D** Top: qPCR analysis of CD274 (PD-L1) expression in 393P (blue) and 344SQ (red) cell lines treated with DMSO or 10 μM AZD6244 for 48 h. Bottom: percent of PD-L1+393P (blue) and 344SQ cells (red) analyzed by flow cytometry following treatment with indicated concentrations of AZD6244 for 48 h. Data are presented as mean values ± SD. n = 3. Data were analyzed using unpaired Students t test. **P < 0.01; ***P < 0.001; ****P < 0.0001. **E** Western blotting of indicated proteins in 393P and 344SQ cells treated with indicated concentrations of AZD6244 for 48 h. **F** Heatmap of RPPA profile showing statistically significant (FDR < 0.05) differentially regulated proteins in 344SQ tumors treated with PD-L1 blocking antibody (black) or IgG isotype control (grey) for 7 weeks. **G** Western blotting of indicated proteins of 344SQ tumors treated with IgG isotype control or PD-L1 blocking antibody for 7 weeks. Number indicates tumor replicate.

with PD-L1 blockade starting at 3 weeks after syngeneic tumor implantation showed a marked synergistic reduction in 344SQ KP lung tumor growth and metastases that lasted for 8–9 weeks, compared with monotherapy or vehicle/isotype controls (Fig. 2a). However, 344SQ tumors ultimately developed resistance to the combinatorial therapy and by ~11–12 weeks produced primary tumor size and metastatic disease (Fig. 2a) that were comparable to untreated controls at 9 weeks. Immune profiling by flow cytometry of tumor tissues (sample gating strategies for immune populations shown in Supplementary Fig. 2) at either the experimental endpoints or following 2 weeks of treatment showed a significant increase in total CD8$^+$ T cells when anti-PD-L1 was administered as a single-agent or in combination with AZD6244 (Fig. 2b, Supplementary Fig. 3a). Furthermore, memory/effector CD8$^+$ T-cell subpopulations were significantly increased, whereas naive and exhausted CD8$^+$ T cells were significantly decreased in the combinatorial treatment group at the point of efficacy (Fig. 2c, Supplementary Fig. 3a). However, despite the transient benefits in the intratumoral T-cell populations, exhausted CD8$^+$ T cells (PD-1$^+$TIM-3$^+$) were significantly increased when tumors eventually developed resistance to combination treatment (Fig. 2c).

Similarly, we treated MEK inhibitor-sensitive, epithelial- 393P KP tumors with combinatorial therapy and observed no significant difference in primary tumor growth rate between single-agent AZD6244 and the combination group (Fig. 2d). Consistent with our prior publication, the epithelial 393P tumors also developed resistance to either the single-agent or combination treatments, but interestingly, the combination group showed a reduction in metastases compared with the AZD6244 monotherapy-resistant group, even at the point (10 weeks) when the primary tumors displayed resistance (Fig. 2d). Immune profiling of 393P tumor tissues at the experimental endpoints or after 2 weeks of treatment showed an increase in total CD8+ T cells only when tumors were treated with AZD6244 monotherapy or in combination with anti-PD-L1 (Fig. 2e and Supplementary Fig. 3b). Again, we only observed a significant increase in memory/effector CD8$^+$ T-cell subpopulations during the responsive period to treatment with AZD6244 or combination therapy (Fig. 2f and Supplementary Fig. 3b). Nanostring analysis on immune markers in 344SQ and 393P tumors in the different treatment groups confirmed our flow cytometry data, showing that treatment of tumors resulted in an increase of T-cell gene signatures. Notably, tumors that were resistant to combination treatment showed a decrease in T-cell gene signatures (Supplementary Fig. 3c, d). Analysis of additional immune cell populations, including CD4$^+$ T-cell subgroups and antigen-presenting cells, in 344SQ and 393P tumors after 2 weeks of treatment did not show statistically significant or consistent changes between the treatment groups (Supplementary Fig. 3e–h).

Our prior work demonstrated that autochthonous mutant KP lung tumors in the genetically engineered mouse model (GEMM) were resistant to AZD6244 monotherapy[7], but were partially responsive to anti-PD-L1 blockade for ~12 weeks before developing resistance[20]. To test out findings from syngeneic tumor models in GEMMs, we treated KP mice with anti-PD-L1 alone or in combination with AZD6244 to determine whether combination therapy could prevent resistant tumor outgrowth. Micro-CT imaging of mice lungs showed an initial reduction in lung tumor growth in mice that received the combination therapy or anti-PD-L1 monotherapy when compared with untreated controls (Fig. 2g). However, lung tumors developed resistance to single-agent and combination therapy after 17 weeks of treatment, consistent with our prior reports (Fig. 2g)[20].

**Combination therapy-resistant syngeneic and autochthonous tumors have increased levels of Th17 CD4$^+$ T cells**. Although combinatorial therapy in 344SQ and 393P tumors exhibited differences in tumor growth and metastatic phenotypes, both tumor types exhibited similar changes in CD8$^+$ T-cell patterns at the point of sensitivity and resistance to the respective drug treatments, suggesting a mutual secondary immune cell infiltrate that promotes resistance following CD8$^+$ T-cell infiltration. To identify specific immune cell populations and cytokines upregulated in drug-resistant 393P and 344SQ tumors, we performed a qPCR array analyzing multiple cytokines, immune-associated receptors, and transcription factors. Data from the immune qPCR array of 344SQ and 393P tumors treated with individual or combinatorial therapies described in Fig. 2a, d showed a consistent upregulation of genes associated with Th17 CD4$^+$ T cells (*IL23, IL17, IL22,* and *RORγt*) in resistant 393P and 344SQ tumors (Fig. 3a, b; Supplementary Fig. 4a, b). Th17 cells are a subset of pro-inflammatory CD4$^+$ T cells that express RORγt and secrete the tumor-promoting cytokines IL-17 and IL-22[16,21–29]. Nanostring analysis comparing 344SQ tumors that were responsive and resistant to combinatorial treatment confirmed our qPCR array, showing a significant upregulation of *IL-17F* in resistant 344SQ tumors (Supplementary Fig. 4c). Flow cytometry analysis of 344SQ and 393P tumor tissues showed an increase in CCR5$^+$, CCR6$^+$, and IL-17$^+$RORγt$^+$ CD4$^+$ T cells in tumors that developed resistance to therapy (Fig. 3c, d; Supplementary Fig. 4d). Immunohistochemistry (IHC) stains for RORγt in autochthonous KP lung tumors treated with anti-PD-L1 or combination therapy from Fig. 2g showed an increase in RORγt$^+$ tumor infiltrating cells in KP tumors resistant to either treatment (Fig. 3e).

**MEK inhibition contributes to the differentiation of Th17 cells, which secrete IL-17 and IL-22, promoting tumor cell drug resistance and invasiveness**. Since Th17 cells typically require TGF-β, IL-6, and IL-23 stimulation for differentiation[30], we next wanted to determine whether MEK inhibition of lung cancer cells contributed to Th17 cell differentiation. First, we treated murine KP lung cancer cells with AZD6244, and observed an increase in *TGF-β, IL-6,* and *IL-23* expression following MEK inhibition (Fig. 4a). A similar trend was found in human NSCLC cell lines H358 and H1299 treated in vitro with AZD6244 (Supplementary Fig. 5a). In addition, AZD6244 treatment of human lung cancer cells showed a consistent upregulation of Th17-associated genes (*TGF-β, IL6, IL23, IL17, IL22, IL1b,* and *RORγt*) (Supplementary Fig. 5a). Next, we co-cultured 393P and 344SQ KP cells with anti-CD3/CD28-activated splenocytes, treated the co-culture with AZD6244, and analyzed splenocyte populations for Th17 cells. In vitro co-culture assays showed an increase in CCR5$^+$RORγt$^+$CD4$^+$ T cells with no significant changes in total CD4$^+$ T cells when 393P or 344SQ cells were treated with AZD6244 (Fig. 4b). To determine whether Th17 cells from co-cultures were secreting IL-17, we co-cultured 344SQ cells with anti-CD3/CD28-activated splenocytes, treated cells with AZD6244, anti-PD-L1, or both in combination, and analyzed secreted IL-17 levels in the conditioned media by ELISA. Treatment of 344SQ cells alone did not show a significant change in secreted IL-17 regardless of treatment (Fig. 4c). However, co-culture of splenocytes with 344SQ cells showed detectable levels of IL-17 in the conditioned media, which was further increased (approximately twofold) when cells were treated with the combination therapy (Fig. 4c). Next, we tested whether IL-17 and IL-22 cytokines secreted by Th17 cells promoted lung cancer cell drug resistance and invasion. 393P and 344SQ cells treated with recombinant IL-17 and IL-22 were more invasive through matrigel-coated transwell

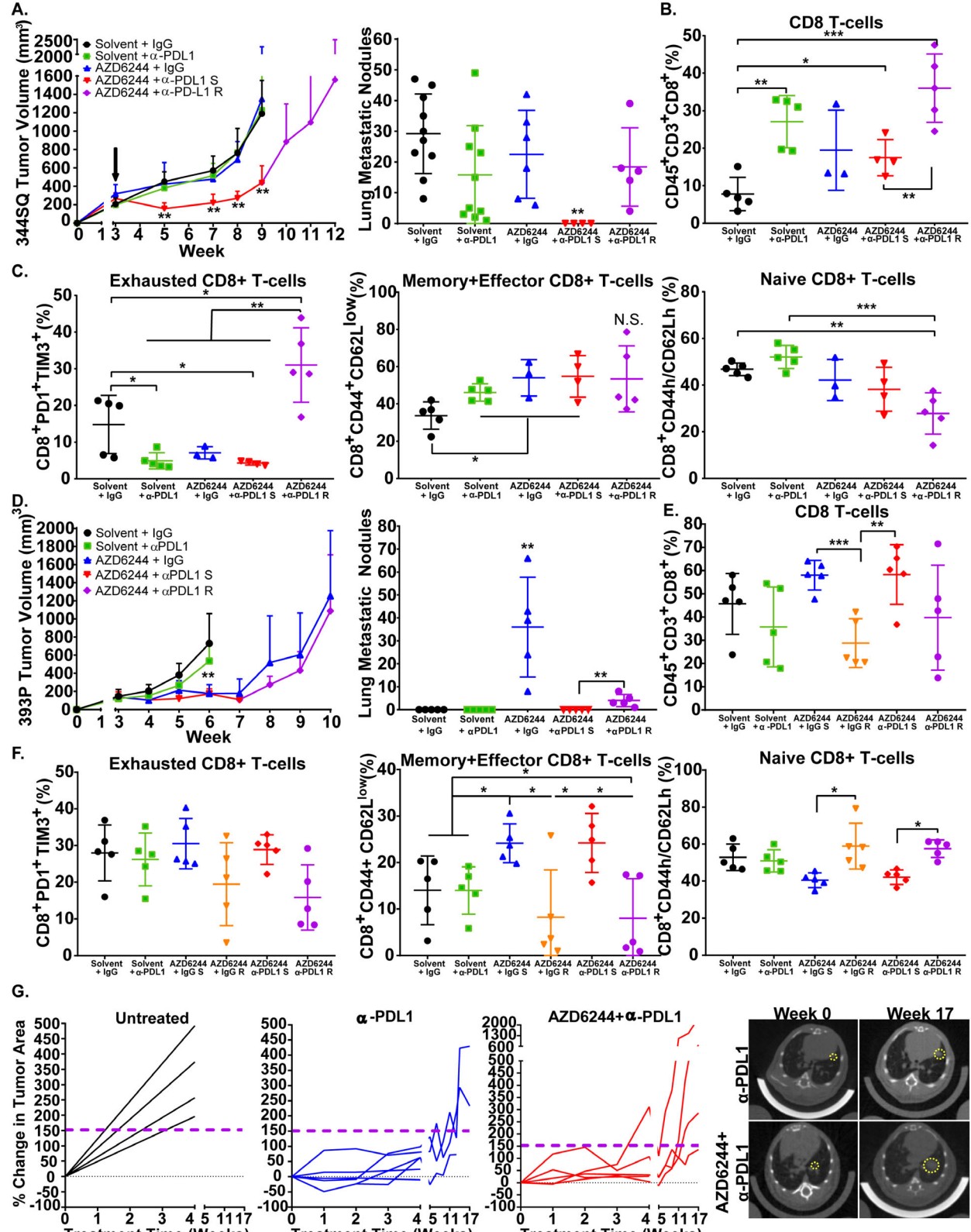

inserts (Fig. 4d) and became less responsive to in vitro AZD6244 MEK inhibition (Fig. 4e).

Although Th17-associated cytokines promoted MEK inhibitor resistance and cancer cell invasion in 393P and 344SQ cells, 344SQ tumors were more metastatic compared with 393P tumors after acquiring resistance to combination therapy, suggesting a differential response to anti-PD-L1. To that end, western blot analysis of immunosuppressive molecules showed that 393P cells upregulated PD-L1 protein levels when cells were treated with IL-17 or IL-22. Conversely, although 344SQ cells had higher basal levels of PD-L1 as previously described[9], stimulation of 344SQ cells with IL-17 decreased PD-L1 expression but dramatically

**Fig. 2 MEK inhibition in combination with PD-L1 blockade initially reduces lung tumor growth and metastasis, but ultimately develops resistance to therapy. A** Left: in vivo tumor volume measurements at indicated time points for 344SQ subcutaneous tumors in syngeneic wild-type mice after daily treatment with solvent (black), 25 mg/kg AZD6244 (blue), or weekly treatment with 200 μg PD-L1 blocking antibody (green) as single-agents or in combination. Treatment start time denoted by black arrow, combination sensitive group denoted by (S) (red), combination resistant group denoted by (R) (purple). Right: quantification of lung metastatic surface nodules in the indicated experimental groups at the endpoint of treatments. Data are presented as mean values ± SD. $N = 10$ (Solvent+IgG), $n = 10$ (Solvent+anti-PD-L1), $n = 6$ (AZD6244 + IgG), $n = 4$ (AZD6244 + anti-PD-L1 S), and $n = 5$ (AZD6244 + anti-PD-L1 R) mice. Data were analyzed using unpaired Students' $t$ test. **$P < 0.01$. **B** Percent of CD45+CD3+ total CD8+ T cells in 344SQ tumors with indicated treatment groups at endpoint of experiment in **A**. Data are presented as mean values ± SD. $N = 3$–5 mice per group. Data were analyzed using unpaired Students $t$ test. *$P < 0.05$; **$P < 0.01$; ***$P < 0.001$. **C** Left: percent of PD1+TIM3+ exhausted CD8+ T cells. Middle: percent of CD44+CD62L− memory/effector CD8+ T cells. Right: percent of CD44+CD62+ naive CD8+ T cells. All populations gated from CD8+ T cells in **B**. Data are presented as mean values ± SD. $N = 3$–5 mice per group. Data were analyzed using unpaired Students $t$ test. N.S. not significant; *$P < 0.05$; **$P < 0.01$; ***$P < 0.001$. **D** Left: in vivo tumor volume measurements at indicated time points for 393P subcutaneous tumors in syngeneic wild-type mice after daily treatment with solvent (black), 25 mg/kg AZD6244 (blue), or weekly treatment with 200 μg PD-L1-blocking antibody (green) as single-agents or in combination. Treatment start time denoted by the black arrow. Treatment start time denoted by black arrow, combination sensitive group denoted by (S) (red), combination resistant group denoted by (R) (purple). Right: quantification of lung metastatic surface nodules in the indicated experimental groups at the endpoint of treatments. Data are presented as mean values ± SD. $n = 5$ mice per group. Data were analyzed using unpaired Students' $t$ test. **$P < 0.01$. **E** Percent of CD45+CD3+ total CD8+ T cells in 393P tumors with indicated treatment groups at endpoint of experiment in **D**. Data are presented as mean values ± SD. $N = 5$ mice per group. Data were analyzed using unpaired Students $t$ test. **$P < 0.01$; ***$P < 0.001$. **F** Left: percent of PD1+TIM3+ exhausted CD8+ T cells. Middle: percent of CD44+ CD62L− memory/effector CD8+ T cells. Right: percent of CD44+CD62+ naive CD8+ T cells. All populations gated from CD8+ T cells in **E**. Data are presented as mean values ± SD. $N = 5$ mice per group. Data were analyzed using unpaired Students $t$ test. *$P < 0.05$. **G** Left: percent change in overall lung tumor area of age-matched $Kras^{G12D};p53^{-/-}$ (KP) at indicated time points following weekly treatment with 200 μg PD-L1 blocking antibody monotherapy (blue) or in combination (red) with daily treatments of 25 mg/kg AZD6244, as assessed by micro-CT imaging of mouse lungs. Right: representative cross-sectional micro-CT images of KP mouse lungs before indicated treatment (Week 0) and treatment endpoint (Week 17). Yellow circles outline representative target lesions. Each line represents an individual mouse.

increased CD38, which promotes an immunosuppressive tumor microenvironment and resistance to PD-(L)1 immune checkpoint blockade[20]. In addition, western blots also showed an increase in p-Stat3 when 393P cells were treated with IL-17 or IL-22 individually or in combination, whereas 344SQ cells had higher basal levels of p-Stat3 and increased p-Stat3 signaling only with IL-22 stimulation (Fig. 4f).

To address the differential response to IL-17 and IL-22 stimulation between 393P and 344SQ cells, we analyzed expression of IL-17 and IL-22 receptors in a panel of mouse KP cell lines. Although *IL17RA* and *IL17RC* expression was similar across all cell lines (Supplementary Fig. 5b), we observed a marked increase in *IL22RA1* for all metastatic, mesenchymal KP cells (Fig. 4g) compared with the non-metastatic, epithelial cells. In addition, levels of *IL22RA1* were elevated in 393P cells following ZEB1 overexpression or acquisition of resistance to AZD6244 (Supplementary Fig. 5c). Conversely, overexpression of miR-200 in mesenchymal 344SQ cells significantly reduced *IL22RA1* expression, indicating a strong correlation between EMT and IL22RA1 expression (Supplementary Fig. 5c). *IL22RA1* could also be induced in 393P cells following stimulation with recombinant IL-17A (Supplementary Fig. 5c). Our findings suggest a working model in which MEK inhibition in lung cancer cells induces secretion of the Th17 differentiating cytokines IL-6, TGF-β, and IL-23 (Fig. 4h). Infiltrating Th17 cells then secrete IL-17 and IL-22 to promote primary tumor resistance to combination therapy (Fig. 4h).

**Th17 gene signatures predict response and overall survival in cancer patients treated with PD-1 blockade therapy.** To evaluate the clinical relevance of our findings, we analyzed Th17-associated gene signatures in RNA-seq data sets from patients treated with anti-PD-1 to predict response to immune checkpoint blockade therapies. First, we analyzed publicly available, published RNA-seq data from pre-treatment NSCLC patient tumors[31]. Expression of Th17-associated genes (*IL17RA, IL17RC, IL22RA1, RORC, CCR5, CCR6, IL6, IL23, TGFB1*) in pre-treatment NSCLC patient samples revealed that upregulation of *IL17RC* mRNA correlated with the progressive disease to anti-

PD-1 (Fig. 5a) (12 patients defined as progressive disease and 4 patients defined as partial response). In addition, patients with higher *IL17RC* expression showed poorer overall and progression-free survival, though differences were not statistically significant owing to low sample size (Supplementary Fig. 6a, b). Although analysis of other Th17-associated genes did not show significant differences in anti-PD-1 response or survival, our pre-clinical data suggest that upregulation of Th17 cells in tumors occurred only after administering treatment. Unfortunately, data sets of matched pre- and post-treatment samples for lung cancer patients treated with immunotherapies alone or in combination with MEK inhibitors are not available, therefore, we analyzed a publically available, published data set from melanoma patients treated with anti-PD-1 therapies[32]. Consistent with NSCLC patient data sets, almost half of melanoma patients possess activating mutations in the MAPK pathway, including *KRAS* and *BRAF* (Supplementary Fig. 6c). Similarly, expression analysis of Th17-associated genes in pre-treatment melanoma patient samples revealed that upregulation of *IL17RC* mRNA correlated with progressive or stable disease, whereas lower *IL17RC* mRNA correlated with partial or complete response to anti-PD-1 (Fig. 5b). Gene expression comparisons between pre- and on-treatment melanoma patient samples showed a significant overall increase in *IL17RA, TGFB1*, and *CCR5* when patients received anti-PD-1 (Fig. 5c). Because patients showed variable changes in gene expression for multiple markers, analysis of net changes (directional change in expression between pre- and on-treatment biopsies) in Th17-associated gene expression predicted significant improvement in overall survival in patients who showed decreased expression in *RORγt, IL17RA, TGFB1*, and *CCR5* (Fig. 5d). There were no significant differences in overall survival in patients with net changes in other genes associated with CD4 subpopulations (Supplementary Fig. 6d). Our analyses of clinical data corroborate our experimental findings and identify Th17-related genes as promising markers to predict patient outcome following immune checkpoint blockade.

**Neutralizing IL-17 abrogates resistance to combination MEK inhibition and PD-L1 blockade therapy.** To test the in vivo

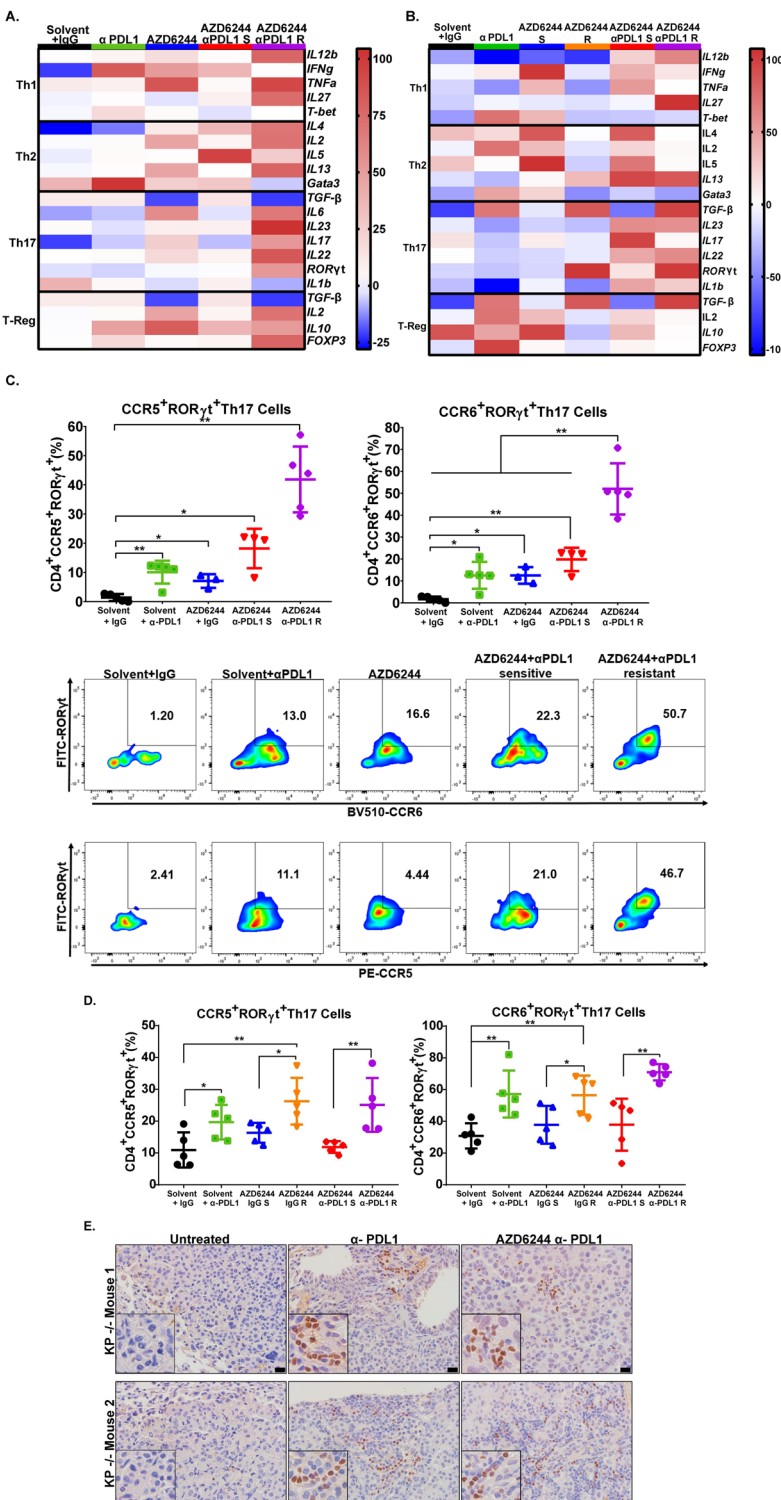

**Fig. 3 Combination therapy-resistant tumors have increased levels of Th17 CD4+ T cells. A** Cytokine qPCR array heatmap of 344SQ tumors from the experiment in Fig. 2A. Solvent + IgG (black), 25 mg/kg AZD6244 (blue) or weekly treatment with 200 μg PD-L1 blocking antibody (green) as single-agents or in combination sensitive group denoted by (S) (red), combination resistant group denoted by (R) (purple). **B** Cytokine qPCR array heatmap of 393P tumors from the experiment in Fig. 2D. **C** Top: percent of CCR5+RORγt+ and CCR6+RORγt+Th17 CD4+ T cells in 344SQ tumors from treatment experiment in Fig. 2A. All populations gated from total CD4+ T cells in Supplementary Fig. S3D. Bottom: representative dot plot for CCR5+RORγt+ and CCR6+RORγt+Th17 CD4+ T cells. Data are presented as mean values ± SD. $N = 3$–5 mice per group. Data were analyzed using unpaired Students' t test. *$P < 0.05$; **$P < 0.01$. **D** Percent of CCR5+RORγt+ and CCR6+RORγt+Th17 CD4+ T cells in 393P tumors from treatment experiment in Fig. 2D. Data are presented as mean values ± SD. $N = 3$–5 mice per group. Data were analyzed using unpaired Students' t test. *$P < 0.05$; **$P < 0.01$. **E** RORγt IHC stains of KP mice lung tumors treated weekly with 200 μg of PD-L1-blocking antibody alone or in combination with 25 mg/kg daily treatments of AZD6244 for 17 weeks from the experiment in Fig. 2G. Scale bar = 50 μm.

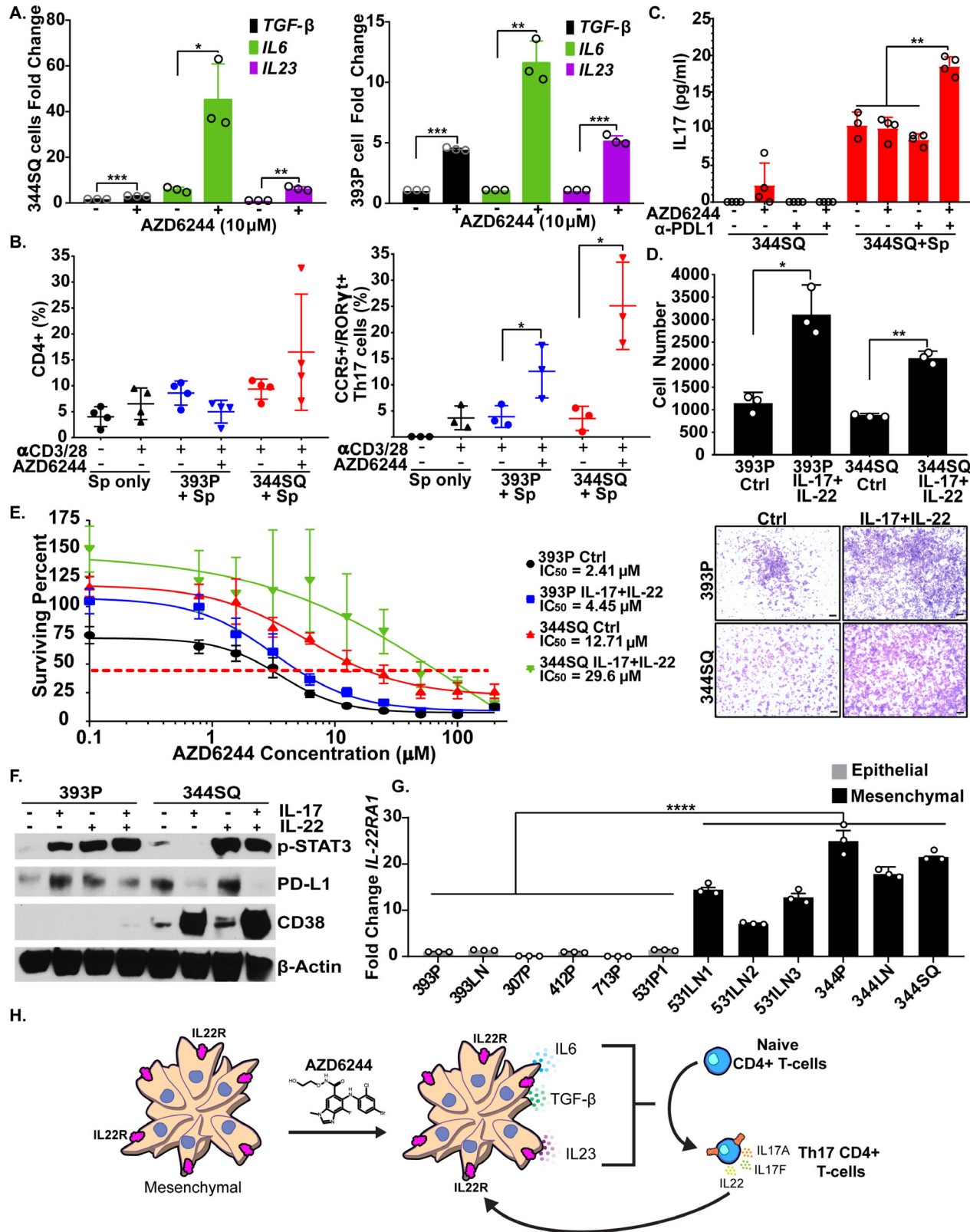

effects of the increased IL-17 production, therapeutic antibody neutralization of IL-17 was combined with MEK inhibition, PD-L1 checkpoint blockade, or MEK inhibitor plus anti-PD-L1 combination in 344SQ tumors starting 3 weeks post implantation in mice. Initial treatment of 344SQ tumors demonstrated that the triple therapy most significantly reduced primary tumor growth

and lung metastasis compared with control, monotherapy, or varying double combination therapy groups after 3 weeks of treatment (Fig. 6a–c). To assess the triple combination therapy in an additional KP lung tumor model, we treated 393P tumors that acquired resistance to selumetinib as previously characterized[7] (393P-AR1) with the identical treatment groups and observed a

**Fig. 4 MEK inhibition contributes to the differentiation of Th17 cells, which secrete IL-17 and IL-22 to promote drug resistance and invasiveness in lung cancer cells. A** QPCR analysis of TGF-β (black), IL-6 (green), and IL-23 (purple) gene expression in 344SQ (left) and 393P (right) murine KP cell lines after treatment with 10 μM AZD6244 for 48 h. Data are presented as mean values ± SD. n = 3. Data were analyzed using unpaired Students' t test. *P < 0.05; **P < 0.01; ***P < 0.001. **B** Left: percent of CD4$^+$ T cells gated from CD45$^+$CD3$^+$ cells in total anti-CD3/CD28-activated splenocytes (Sp) co-cultured with 393P (blue) or 344SQ cells (red) and treated with 10 μM AZD6244 for 96 h. Right: percent of CCR5$^+$RORγt$^+$ Th17 cells gated from total CD4$^+$ T-cell population (Left) following co-culture and treatment conditions previously described. Data are presented as mean values ± SD. n = 3–4. Data were analyzed using unpaired Students' t test. *P < 0.05. **C** IL-17 concentration as measured by ELISA in conditioned media of 344SQ cells cultured alone or in co-culture with splenocytes (Sp), treated with 10 μM AZD6244 and/or 20 μg/ml PD-L1 blocking antibody for 96 h. Data are presented as mean values ± SD. n = 3–4. Data were analyzed using unpaired Students' t test. **P < 0.01. **D** Quantification and representative images of 393P and 344SQ cell invasion through matrigel-coated transwell inserts ± IL-17 and IL-22 cytokine stimulation. Data are presented as mean values ± SD. n = 3. Data were analyzed using unpaired Students' t test. *P < 0.05; **P < 0.01. Scale bar = 200 μm. **E** In vitro cell survival response after 72-hour selumetinib (AZD6244) treatment in 393P and 344SQ cells ± IL-17 and IL-22 cytokine stimulation. Data are presented as mean values ± SD. N = 6–8 replicates per concentration. **F** Western blot of indicated proteins in 393P and 344SQ cells treated with IL-17 and/or IL-22 individually or in combination for 96 h. **G** QPCR analysis of IL22RA1 expression in panel of murine epithelial cells (grey) or mesenchymal cells (black). Data are presented as mean values ± SD. n = 3 technical triplicates. Data were analyzed using unpaired Students t test. ****P < 0.0001. **H** Proposed working model. Initial response to combination therapy promotes the release of IL-6, TGF-β, and IL-23 cytokines by resistant tumor cells. The released cytokines promote the CD4+ T-cell differentiation into Th17 cells, which secrete IL-17 and IL-22 to promote resistance to MEK inhibition and PD-L1 blockade.

significant decrease in tumor growth only with treatment of the triple combination therapy (Fig. 6d, e). To determine whether the triple therapy could sustain long-term reduction in tumor growth and prevent resistant outgrowth, we treated 344SQ tumors for 10 weeks, which showed that triple therapy abrogated resistant tumor outgrowth and metastasis compared with tumors treated with the double combination therapy (Fig. 6f–h).

Immune profiling of the tumors at the endpoint of the experiment showed an increase in total CD8$^+$ and memory/effector CD8$^+$ T cells followed by a decrease in exhausted and naive CD8$^+$ T cells (Fig. 7a). Although total CD4$^+$ T-cell levels did not change (Supplementary Fig. 7a), we observed a consistent increase in CCR6$^+$ and CCR5$^+$RORγt$^+$ Th17 cells in tumors resistant to the double combination treatment, whereas tumors treated with the triple combination therapy prevented the increase in Th17 populations (Fig. 7b). Our findings suggest a working model in which single-agent treatment selects for either a resistant epithelial or mesenchymal population, whereas combinatorial therapy of MEK inhibitor with PD-L1 blockade reduces both epithelial and mesenchymal subpopulations to suppress overall lung tumor growth (Fig. 7c). Although infiltration of IL-17-secreting Th17 cells promotes resistance to the combination therapy, neutralization of IL-17 abrogates resistance, produces sustained activated CD8$^+$ T-cell levels and lung tumor reduction, validating a promising triple combinatorial treatment strategy to overcome single-agent or combination therapy drug resistance in KRAS driven lung cancers (Fig. 7c).

## Discussion

The failure of MEK inhibitors against KRAS mutant lung cancers and the low percentage of sustained durable responses with anti-PD-1/PD-L1 immune checkpoint blockade therapies emphasize the need to understand drug resistance mechanisms to improve patient survival. Here, we identified a reciprocal expression between PD-L1 and MAPK signaling when lung tumors were treated with MEK inhibitor or anti-PD-L1, respectively, validating a promising dual combinatorial therapy approach previously reported in multiple cancer types[13,33–36]. Consistent with previous work[34,37], MAPK inhibition in combination with PD-L1 blockade demonstrated an initial reduction in KP lung tumor growth. However, despite the initial response to the double combination therapy and even anti-PD-L1 monotherapy in our KP pre-clinical GEMMs, lung tumors eventually developed resistance to combination therapy. Furthermore, the syngeneic KP tumor models demonstrated that distinct subsets of lung cancer cells with varying baseline sensitivities to MEK inhibitor also developed

resistance to double combination therapy, suggesting that a mutual factor promotes resistance. Here, we identified a dramatic increase in Th17 CD4$^+$ T cells in resistant tumors in both the KP syngeneic and GEM models. The increase in subpopulations of CD4$^+$ T cells is partially corroborated in melanoma phase 1b studies involving combination MAPK inhibitor and PD-L1 checkpoint blockade[33]. In addition, our data demonstrated that MEK inhibition or PD-L1 blockade alone was sufficient to significantly increase Th17 tumor infiltration. We further demonstrate that IL-17 and IL-22 cytokines secreted by Th17 cells are directly responsible for promoting drug resistance, invasion, and metastasis. Analysis of immune-suppressive markers between different lung cancer models showed differential response to Th17 cytokines. In 393P cells, which express low basal levels of PD-L1, IL-17, and IL-22 stimulation upregulated PD-L1 expression, whereas 344SQ cells, which express high basal levels of PD-L1 and IL-17, specifically suppressed PD-L1 expression but dramatically upregulated CD38 levels. We previously demonstrated that CD38 is crucial for immune evasion and PD-L1 blockade resistance[20]. Here, we demonstrate that infiltration of Th17 cells and stimulation of cancer cells with IL-17 is a mechanism of CD38 upregulation following treatment to promote immunotherapy resistance. In addition, the increased basal levels of Th17 cells in PD-L1$^{low}$ 393P tumors suggest that Th17 cells may be a mechanism for primary immune evasion and tumor formation.

The observation that KL cell lines increase PD-L1 levels is significant as previous reports demonstrate that STK11 (LKB1) co-mutations in KRAS mutant lung tumors promote resistance to anti-PD-(L)1 therapies owing to low expression of PD-L1[38]. Though our pre-clinical findings were contained within the KP syngeneic model, the increase in PD-L1 expression in mutant KRAS;STK11 (KL) lung cancer cell lines suggest that our proposed therapeutic strategy may be extended to KL lung cancer subtypes. Further investigation as to which co-occurring mutations lead to an increase infiltration of Th17 cells in resistant tumors will provide a better understanding of how Th17 cells could potentially promote baseline versus acquired tumor immune evasion.

Although Th17 cells are commonly associated with a pro-inflammatory response[22], analysis of RNA-seq data from melanoma patient biopsies during pre- and on-treatment of PD-1 blockade showed that a net increase in Th17-associated gene signatures correlated with poorer response and predicted poorer overall survival following treatment. These clinical findings corroborate our experimental data and identify Th17 genes as potential markers of resistance.

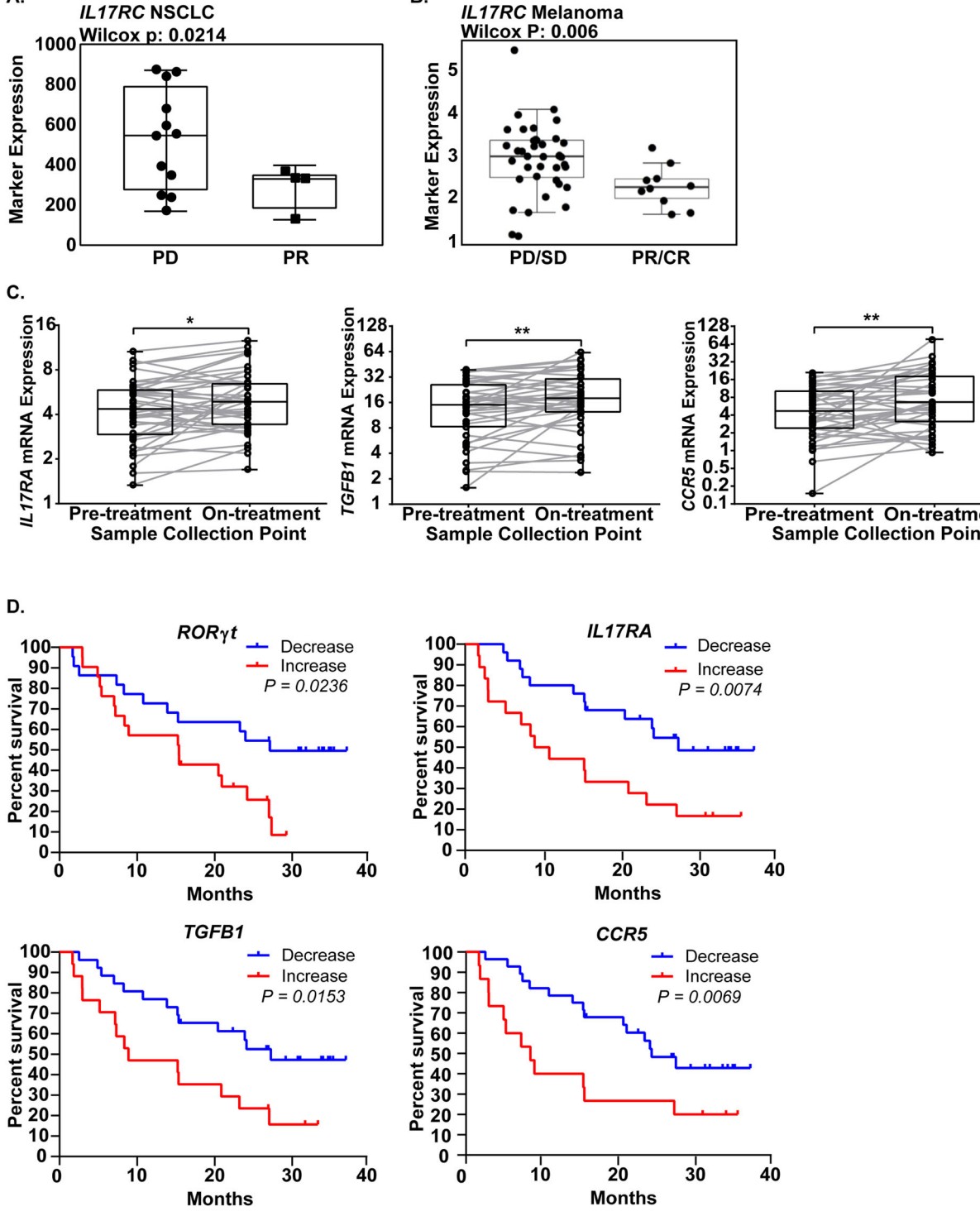

**Fig. 5 Th17 gene signatures correlate with decrease in survival in IO-treated melanoma patients. A** IL17RC mRNA levels in NSCLC patients that exhibited progressive disease (PD) versus partial response (PD) to anti-PD-1 therapy (nivolumab). 12 patients defined as a progressive disease and four patients defined as a partial response. Statistics calculated using two-sided Wilcoxon matched pair rank test with significance at $P < 0.05$. **B** Pre-treatment IL17RC mRNA levels in melanoma patients that exhibited progressive or stable disease (PD/SD) versus partial or complete response (PR/CR) to anti-PD-1 therapy (nivolumab). 36 patients defined as PD/SD and 10 patients defined as PR/CR. Statistics calculated using two-sided Wilcoxon matched pair rank test with significance at $P < 0.05$. **C** Left: IL17RA, middle: TGFb1, and right: CCR5 mRNA expression levels in melanoma patient samples in pre- and on-treatment of nivolumab. Statistical difference was determined using Wilcoxon matched pair rank test. 39 independent patient samples for pre-treatment and on-treatment. Boxplots are shown as the median ± 1 quartile, whiskers extend to an extreme data point. Statistics calculated using two-sided Wilcoxon matched pair rank test with significance at $*P < 0.05$, $**P < 0.01$. **D** Kaplan–Meier curves predicting survival of nivolumab-treated melanoma patients based on net changes in RORγt, IL17RA, TGFB1, and CCR5. Percent survival decreases shown in blue and increase shown in red. Statistical difference was determined using the Log-rank Cox test.

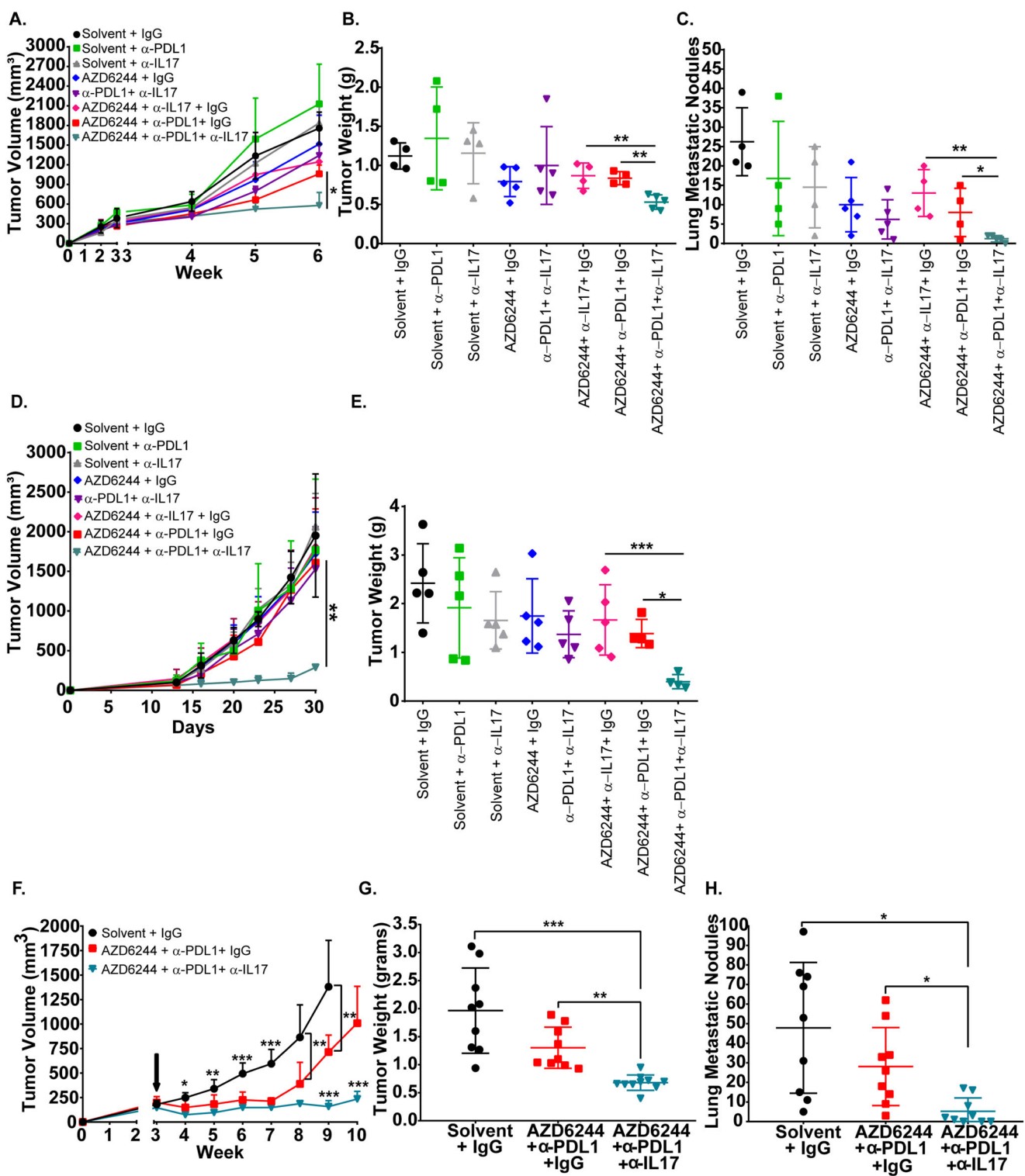

Our identification that the Th17-associated cytokine IL-17 promotes resistance to anti-PD-L1 alone or in combination with MEK inhibition validates a novel triple combinatorial therapeutic strategy for lung cancer patients, consistent with observations that IL-17 is involved in tumor progression[21,25]. Moreover, the increase in cancer cell IL-6 expression from combination therapy as well as the role of Th17 in autoimmunity suggests that IL-17 blockade could potentially ameliorate the observed clinical adverse events owing to immune checkpoint blockade[16,39–41]. A major advantage of our proposed triple combinatorial treatment strategy is the fact that all three targets currently have drugs that have received FDA approval, as IL-17 drugs are FDA approved to treat psoriasis[42]. Moreover, results from our study emphasize the need for accessible lung cancer patient data sets with matched pre- and post-treatment biopsies to predict patient response and identify potential combination therapeutic strategies to improve patient survival.

**Fig. 6 Neutralizing IL-17 abrogates resistance to MEK inhibition and PD-L1 blockade combination therapy. A** In vivo tumor volume measurements at indicated time points for 344SQ subcutaneous tumors in syngeneic wild-type mice after daily treatments with 25 mg/kg AZD6244 in combination with weekly treatments with 200 μg PD-L1 and/or 200 μg IL-17 blocking antibody beginning at week 3 of tumor growth. Solvent + IgG (black), daily 25 mg/kg AZD6244 (blue), weekly treatment with 200 μg PD-L1 blocking antibody (green), weekly treatment with 200 μg IL-17 blocking antibody (grey), combination blocking antibodies 200 μg PD-L1 + 200 μg IL17 (purple), combination 25 mg/kg AZD6244 + weekly 200 μg blocking antibody IL17 (pink), combination 25 mg/kg AZD6244 + weekly 200 μg blocking antibody PD-L1 (red), triple combination 25 mg/kg AZD6244 + weekly 200 μg blocking antibody PD-L1 + 200 μg blocking antibody IL17 (teal). Data are presented as mean values ± SD. $n = 4$–5. Data were analyzed using unpaired Students' $t$ test. *$P < 0.05$. **B, C** Tumor weights **B** and lung metastatic nodules **C** of 344SQ tumors at endpoint of experiment in **A**. Data are presented as mean values ± SD. $n = 4$–5. Data were analyzed using unpaired Students' $t$ test. *$P < 0.05$; **$P < 0.01$. **D** In vivo tumor volume measurements at indicated time points for 393P-AR1 subcutaneous tumors in syngeneic WT mice after daily treatments with 25 mg/kg AZD6244 in combination with weekly treatments with 200 μg PD-L1 and/or 200 μg IL-17 blocking antibody beginning at day 12 of tumor growth. Solvent + IgG (black), daily 25 mg/kg AZD6244 (blue), weekly treatment with 200 μg PD-L1 blocking antibody (green), weekly treatment with 200 μg IL-17 blocking antibody (grey), combination blocking antibodies 200 μg PD-L1 + 200 μg IL17 (purple), combination 25 mg/kg AZD6244 + weekly 200 μg blocking antibody IL17 (pink), combination 25 mg/kg AZD6244 + weekly 200 μg blocking antibody PD-L1 (red), triple combination 25 mg/kg AZD6244 + weekly 200 μg blocking antibody PD-L1 + 200 μg blocking antibody IL17 (teal). Data are presented as mean values ± SD. $n = 4$–5 mice per group. Data were analyzed using unpaired Students' $t$ test. **$P < 0.01$. **E** Tumor weights of 393P-AR1 tumors at endpoint of experiment after indicated treatment combinations from **D**. Data are presented as mean values ± SD. $n = 5$. Data were analyzed using unpaired Students' $t$ test. *$P < 0.05$; ***$P < 0.001$. **F** In vivo tumor volume measurements at indicated time points for 344SQ subcutaneous tumors in syngeneic wild-type mice after daily treatments with 25 mg/kg AZD6244 in combination with weekly treatments with 200 μg PD-L1 blocking antibody or 200 μg IL-17A blocking antibody. Black arrow indicates treatment start time. Solvent + IgG (black), combination 25 mg/kg AZD6244 + weekly 200 μg blocking antibody PD-L1 (red), triple combination 25 mg/kg AZD6244 + weekly 200 μg blocking antibody PD-L1 + 200 μg blocking antibody IL17 (teal). Data are presented as mean values ±SD. $n = 9$ mice per group. Data were analyzed using unpaired Students' $t$ test. *$P < 0.05$; **$P < 0.01$; ***$P < 0.001$. **G, H** Tumor weights **G** and lung metastatic nodules **H** of 344SQ tumors at endpoint of experiment in **F**. Data are presented as mean values ± SD. $n = 9$ mice per group. Data were analyzed using unpaired Students' $t$ test. *$P < 0.05$; *$P < 0.01$; ***$P < 0.001$.

## Methods

**Cell culture and reagents**. Human and murine cell lines were cultured in Roswell Park Memorial Institute Medium 1640 (Gibco) with 10% fetal bovine serum (Gibco). Murine cell lines 344SQ and 393P (Kras $^{LA1/+}$/p53$^{R172HΔg/+}$) were generated from dissociated tumor nodules and seeded on tissue culture plates[6]. Human cell lines were obtained from the American Type Culture Collection. All cell lines were routinely tested for mycoplasma contamination on a monthly basis using the LookOut Mycoplasma PCR Detection Kit (Sigma). Any cells yielding positive mycoplasma results were immediately discarded.

**Tumor models and in vivo treatments**. All animal experiments were reviewed and approved by the Institutional Animal Care and Use Committee at The University of Texas MD Anderson Cancer Center. All authors have complied with all relevant ethical regulations for animal testing and research. Syngeneic tumor assays for male or female wild-type 129/sv mice were used between the ages of 6–8 weeks at the start of each experiment. Mice were housed in ventilated cage enclosures in an environment maintained at 45% humidity with ambient temperatures range between 70 °F and 74 °F and 12-h-light/dark cycles. Cancer cells were injected subcutaneously into the mouse flank (unless otherwise noted $1 \times 10^6$ cells in 100 μl of Dulbecco's phosphate-buffered saline). Tumor size was calculated using the formula ½ (length × width$^2$) at the time point indicated. Surface lung metastatic nodules were counted and lungs were then perfused, fixed in 10% formalin, and processed for sectioning[19]. Treatment was started at week 3 or when tumors were an average of 150–200 mm$^3$ as measured by a digital caliper. Mice were randomized before receiving vehicle control, treatment, or combination therapy. Mice were treated with antibodies 200 μg of anti-PD-L1 (clone 10 F.9G2), anti-PD1 (clone RMP1-14), or anti-IL17A (clone 17F3) per mouse or their IgG controls via i. p. injection once per week for the indicated time. Selumetinib (SelleckChem) (4% dimethyl sulfoxide, 30% PEG300, 5% Tween80) or vehicle control was administered daily by oral gavage at 25 mg/kg mouse weight. Tumor sizes were measured weekly after treatment.

The Kras$^{LSL-G12D/+}$;p53$^{fl/fl}$ (KP$^{−/−}$) adeno-Cre inducible mouse model (129/sv background) of lung adenocarcinoma was used[43]. Male or female mice were infected with virus by intratracheal intubation at ~3 months of age. Drug treatment of these mice with selumetinib and/or anti-PD-L1 antibody was started at 3 months post infection when the presence of tumors was confirmed by micro-CT imaging. All mice were genotyped to determine the mutational status by tail snips 2 weeks after birth. Transverse cross-sectional CT images were analyzed for largest tumor area quantification. Tumor diameters were measured using image-J.

**Co-culture assay**. Spleens were isolated from tumor-free male or female 129/sv mice of at least 3 months of age. Spleen was placed on a 0.45 μM strainer and a plunger was used to press the spleen through the strainer. Splenocytes collected in the flow through were further washed with fluorescence-activated cell sorting (FACS) buffer and RBC lysis was performed following manufacturer recommendations (Biolegend). Cell count and viability check using trypan blue was performed before culturing splenocytes with tumor cells. Viable splenocytes were co-cultured with the indicated tumor cell line at a ratio of 1:20 (tumor cell line:

splenocyte) in the presence of anti-CD3/anti-CD28 (5 μg/ml) (Life technologies) for 96 h at 37°C, 5% CO$_2$. Where indicated AZD6244 (10 μM) and/or anti-PDL1 (20 μg/ml) were added at the time of cell seeding.

**Invasion/migration assays**. Transwell migration of 8 μM inserts (BD Biosciences) and invasion (BD-Bioscience) assays was performed for 16 h[6]. Inserts were imaged using crystal violet solution and migratory or invasive cells analyzed on an Olympus IX73 microscope and counted using Image-J software. Five microscopy fields were taken per transwell insert for quantitation.

**Flow cytometry analysis**. Tumors were processed following the MACS (Miltenyi) mouse tumor dissociation kit. Mechanical dissociation using a gentleMACS Octo dissociator (Miltenyi) was performed followed with enzymatic digestion with collagenase I (0.05% w/v, Sigma), DNase type IV (30 U/ml, Sigma), and hyaluronidase type V (0.01% w/v, Sigma) for 40 min with rocking at 37°C. Tumor samples were mechanically dissociated again and passed through a 70 μm filter before being stained with fluorochrome-conjugated antibodies in FACS buffer. RBC lysis (Biolegend) was performed on both single-cell tumor and splenocytes samples following manufacturer recommendation. Cells were stained for surface markers using fluorochrome-conjugated anti-mouse antibodies (CD45, CD3, CD8, CD4) for 1 h at room temperature. Ghost aqua BV510 (Tonobo) was used to stain dead cells. Cells were fixed using 1% paraformaldehyde at room temperature for 15 min, then washed twice with perm/wash buffer (Biolegend). Cells were stained for intracellular antibodies at room temperature for 1 h. Cells were filtered and analyzed on either a BD Canto II (BD Biosciences, San Jose CA), or BD LSR Fortessa (BD Biosciences) and analyzed using FlowJo software (v.10.5.3 & v.10.6.1 Tree Star). For single-color compensation, ultracomp eBeads compensation beads (Thermo Fisher) were used and stained with a single fluorescent-conjugated antibody according to the manufacturer's instructions. Compensation was calculated automatically using BD FACSDiva v8.0.1. The list and dilution of antibodies used are found in Supplementary Table 1.

**Protein isolation and western blot analysis**. Total cell lysates were obtained from human or murine cell lines[7]. Protein was extracted from flash-frozen tumors using a homogenizer (Fisher Scientific) in lysis buffer (1% Triton X-100, 50 mM HEPES, 150 mM NaCl, 1.5 mM MgCl$_2$) on ice. The samples were pulsed for 5 seconds, until the tissue was disintegrated and then incubated on ice for 30 min. In all, 20 μg of lysate was loaded onto a sodium dodecyl sulphate–polyacrylamide gel electrophoresis gel, electrophoresed, and transferred to a nitrocellulose membrane. After blocking with 5% fat-free dry milk (Bio-rad) membranes were probed with primary antibodies listed in Supplementary Table 1 overnight at 4°C. Horseradish peroxidase (HRP)-conjugated secondary antibodies were added to membranes and incubated for 1 h. The Pierce ECL western blotting substrate (Thermo Fisher) was used.

**RNA isolation and real-time qPCR analysis**. RNA from tumors was isolated using the mirVana isolation kit (Thermo Fisher) per manufacturer's

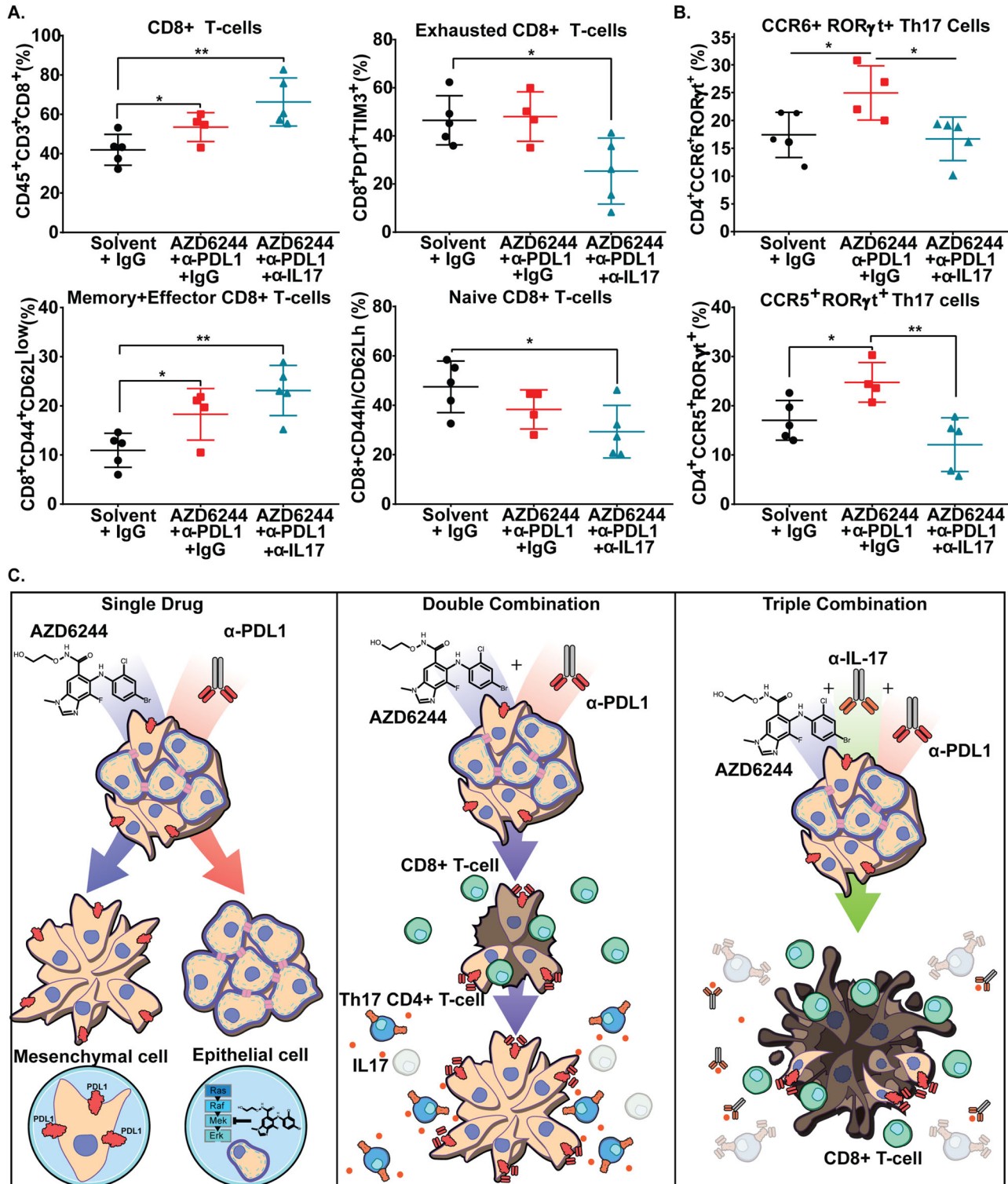

**Fig. 7 Neutralizing IL-17 promotes pro-inflammatory tumor immune microenvironment. A** Percent of CD45+CD3+ total CD8+ T cells and percentage of exhausted, memory/effector, and naive CD8+ T-cell subpopulations gated by the indicated markers from 344SQ tumors from the experiment in Fig. 6F. Data are presented as mean values ± SD. $n = 4$–5. Data were analyzed using unpaired Students' $t$ test. *$P < 0.05$; **$P < 0.01$. **B** Percent of CCR6+ and CCR5+RORγt+ Th17 cells gated from total CD4+ T-cell population. Data are presented as mean values ± SD. $n = 4$–5. Data were analyzed using unpaired Students' $t$ test. *$P < 0.05$; **$P < 0.01$. **C** Proposed working model of MEK inhibitor and anti-PD-L1 combinatorial drug resistance. Initial response to combination therapy promotes CD4+ T-cell differentiation into Th17 cells, which secrete IL-17 and IL-22 to promote resistance to MEK inhibition and PD-L1 blockade. Neutralization of IL-17 in combination with MEK inhibition and PD-L1 blockade abrogates resistant tumor outgrowth.

recommendations. RNA from cell lines was isolated using TRIzol reagent (Thermo Fisher). After quantification of RNA using an Epoch plate reader Take3 plate 260/280 nm (BioTek), 2 µg of RNA was used for reverse transcription using qSCRIPT master mix (Quanta biosciences). QPCR were performed using SYBR green PCR master mix (Thermo Fisher). Primers specific for each gene measured are listed in Supplementary Table 2. L32 was used to normalize expression across all samples. QPCR reactions were performed on the 7500 Fast Real-Time PCR system (Applied Biosystems), CFX96 Real-Time PCR detection system (Bio-Rad Laboratories), or CFX384 Real-Time PCR detection system (Bio-Rad Laboratories).

**Reverse phase protein array**. Tumor tissues were mechanically homogenized in lysis buffer (1% Triton X-100, 50 mM HEPES [pH 7.4], 150 mM NaCl, 1.5 mM MgCl$_2$, 1 mM EGTA, 100 mM NaF, 10 mM NaPPi, 10% glycerol, 1 mM phenylmethylsulfonyl fluoride, 1 mM Na$_3$VO$_4$, with protease and phosphoprotease inhibitors from Roche), incubated on ice for 20 min, centrifuged at $20,000 \times g$ for 10 min, and collected for supernatant. Protein concentration was measured using the Pierce BCA Protein Assay Kit (Thermo Fisher Scientific), and protein samples were prepared to a final concentration of 1 µg/µl after mixing with 4× SDS sample buffer (40% glycerol, 8% SDS, 0.25 M Tris-HCl pH 6.8, 10% 2-mercaptoethanol) to produce a 1× SDS sample buffer solution. Protein samples were then boiled at 100°C for 5 min and stored at −80 °C for RPPA processing described here (https://www.mdanderson.org/research/research-resources/core-facilities/functional-proteomics-rppa-core/rppa-process.html). For analysis of RPPA data, a linear mixed model was applied to compare protein expression on a protein-by-protein basis between epithelial and mesenchymal groups; the model includes cell line effects as a random effect factor. The resulting $p$ values were modeled by a Beta-Uniform Model. To identify protein markers differentially regulated in epithelial and mesenchymal phenotypes, we used an FDR of 0.05 as the cutoff. The heatmap was generated based on mean adjusted expression for each cell line. The Pearson correlation was used for distance matrix calculation, and Ward method was applied as a linkage rule for the hierarchical clustering.

**NanoString**. 129/sv mice implanted with 344SQ or 393P tumors were treated with anti-PD-L1 (200 µg/mouse) once per week i.p. and/or selumetinib (10 mg/kg) for 14 days. Tumors were harvested and flash-frozen, total RNA isolated using the mirVana isolation kit (Thermo Fisher). Gene expression analysis was done using a custom mouse tumor microenvironment panel. A total of 100 ng of total RNA in a final volume of 5 µl was mixed with capture probes and reporter probes tagged with a fluorescent barcode from the custom gene expression code set. Probes and targets were hybridized at 65°C for 12–24 h. Hybridized samples were run on the NanoString nCounter preparation station following manufacturer recommendations. The samples were run on maximum scan resolution on the nCounter Digital Analyzer. Data were analyzed using nSolver Analysis Software. Additional statistical analysis on NanoString nCounter data was conducted as described in[20], using R v3.4.2 & v3.5.1 (R Core Team, 2016). Samples were scaled by the geometric mean of the positive spike-in RNA hybridization controls, as described in the nCounter data analysis manual (NanoString Technologies, Inc., 2011). The expression of each endogenous gene was then tested against the detected expression of all negative control genes using one-sided, two-sample $t$ tests. Genes showing greater expression than the negative control genes with $p < 0.001$ were included in further analysis, whereas others were omitted. Housekeeping gene normalization was applied using the same geometric mean scaling, with a housekeeping gene set identified as most stable in a larger data set by the method of ref. [44]. This set consisted of Alas1, Abcf1, Tbp, Ppia, and Tubb5. Differential expression analysis was conducted on the data after log2 transformation, comparing treatment vs. IgG using the Empirical Bayes method in limma[45,46].

**Immunohistochemistry**. Formalin-fixed paraffin-embedded tumor or lung tissue were cut to 4 µm sections and IHC was performed as follows: 3% hydrogen peroxide for 10 min, 5% goat serum blocking for 1 h, tissues probed with primary antibody listed in Supplementary Table 1 overnight in 4 °C, secondary streptavidin-conjugated secondary antibody was probed for 1 h, incubation with biotinylated HRP for 30 min, 3,3′-diaminobenzidine reagent (Dako) was used for signal development (~5 min). Slides were counter-stained with Harris Hematoxylin (Thermo Fisher) and further dehydrated and mounted. IHC was performed following Peng et al.[7]. Digital images of stained slides were acquired using an Aperio slide scanner AT2 (Leica Biosystems). The Aperio imagescope software version 12.3.3 was used for further analysis.

**Analysis of RNA sequencing data**. FPKM RNA sequence data previously published from Riaz et al.[32] was downloaded from GEO (GSE91061) for melanoma patients. RNA sequence data was also used from Cho et al.[31] and was downloaded from GEO (GSE126044) for NSCLC patients. Patient samples with 0 sequencing reads were excluded from the analysis. Survival analysis was performed using the Kaplan–Meier method and log-rank test was used to determine statistical significance. Comparison of gene expression values between response groups was performed using the Wilcoxon matched-pairs signed-rank test. The number of patients from GSE91061 for each comparison is further described in the table below.

| | Increase | Decrease | Gene |
|---|---|---|---|
| No. of patients | 17 | 11 | RORgt |
| Median survival time (days) | 67.9 | 119.6 | |
| No. of patients | 15 | 12 | IL17RA |
| Median survival time (days) | 43.45 | 120.6 | |
| No. of patients | 14 | 13 | TGFb1 |
| Median survival time (days) | 39.3 | 120.6 | |
| No. of patients | 12 | 15 | CCR5 |
| Median survival time (days) | 37 | 106.7 | |

**Statistics**. All statistical analyses were performed using GraphPad Prism software version 8.0.0. Unless otherwise noted, a one-way analysis of variance post hoc Tukey test was used for multi-group comparisons while unpaired Students' $t$ test (two-tailed) was performed for two-group comparisons. A $p$ value of <0.05 was considered statistically significant. In vitro and in vivo experiments were repeated at least twice independently with similar results.

**Reporting summary**. Further information on research design is available in the Nature Research Reporting Summary linked to this article.

## Data availability
The RNA sequence data that support the findings of this study are available in Gene Expression Omnibus (GEO) with the following accession numbers GSE126044[31] (https://doi.org/10.1038/s12276-020-00493-8) (https://www.ncbi.nlm.nih.gov/geo/query/acc.cgi) and GSE91061[32] (https://doi.org/10.1016/j.cell.2017.09.028).

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

## Acknowledgements

This study made use of the Research Histology, Pathology, and Imaging Core, supported by P30 CA16672 DHHS/NCI Cancer Center Support Grant (CCSG). The Flow Cytometry Lab South Campus Core facility at MDACC is supported by CCSG NCI P30 CA16672. The authors declare that all other data and source data file supporting findings in this study are available within the paper. This work was supported by NIH R37CA214609, CPRIT-MIRA RP160652-P3, and Rexanna's Foundation for Fighting Lung Cancer to D.L.G., LUNGevity Foundation award to D.L.G. & L.A.B, and F32CA239292 to J.M.K. The work was also supported by the generous philanthropic contributions to The University of Texas MD Anderson Lung Cancer Moon Shots Program. Source data are provided with this paper.

## Author contributions

Conceptualization, D.H.P., B.L.R., and D.L.G.; investigation, C.A.G., L.D., J.M.K, J.O.K., J.W.; writing, B.L.R, D.H.P., D.L.G.; resources, A.P., P.O.G., J.J.F, L.G, L.C., L.A.B.; supervision, D.L.G.

## Competing interests

D.L.G. declares advisory board work for Janssen, AstraZeneca, GlaxoSmithKline, and Sanofi. D.L.G. receives research grant funding from AstraZeneca, Janssen, Astellas, Ribon Therapeutics, NGM Therapeutics, and Takeda. L.A.B. declares consulting work for AstraZeneca, AbbVie, GenMab, BergenBio, Pharma Mar, SA. L.A.B. receives research grant funding from AbbVie, AstraZeneca, GenMab, Tolero Pharmaceuticals. All other authors declare that they have no competing interests.
