## [Peer Review File · Nature Communications]

Reviewers' Comments:

Reviewer #1:

Remarks to the Author:

The authors here correctly point out two difficult facts in the treatment of NSCLC: 1) single agent MEK inhibition is for the most part not effective in the treatment of KRAS mutant NSCLC and 2) while immune checkpoint inhibitors have been a great development in the field, most patients will not respond to therapy. Improving response to immune checkpoint inhibitors is an important effort in the field and one certainly worthy of exploration as a topic.

Using established mouse KP model for NSCLC (KRAS mutant/TP53 mutant), that treatment with single agent MEK inhibitor leads to increased PD-L1 expression and treatment with a PD-(L)1 antibody leads to increase in MAPK signaling. Therefore, this is a potentially attractive target for combination therapy.

Shown here is that combination therapy with MEK and PD-(L) inhibition is synergistic initially but only briefly. These experiments are important ones, and clinical trials to date combining PD-L1 and MEK inhibition have not been exciting in terms of results, although many trials have been conducted in an unselected patient population.

The authors went on to evaluate potential mechanisms of resistance examining the immune infiltrate in these tumors. In the setting of resistance, the authors observed and upregulation of Th17 CD4+ T-cells are identified which secrete IL-17 and IL-22

IL-17 + MEK + PD-L1 inhibition is synergistic. This is demonstrated only in the 334SQ tumors. No experiments are shown in the 393P tumors which would be helpful to know if this combination is effective in multiple models or particular to the 334SQ tumor model.

The finds the authors describe are potentially clinically interesting, given that IL-17A antibodies such as the ones used in this manuscript, are already FDA approved and could theoretically be used in a clinical trial.

Comments to authors:

- Is the combination shown here specifically only to the KP model (KRAS/PD53 co-mutation) or do you anticipate to be generalizable to other common co-mutation patterns which may have different responses to immunotherapy (KRAS/LKB1 mutant for example).

- You report that in patients with melanoma increase in IL-17 while on treatment was associated with shorter OS. It is unfortunate that this analysis was performed in melanoma patients and not NSCLC patients. Although it is understandable that there is not a comparison cohort with NSCLC patients with pre and on-treatment tissue analysis available, is baseline pre-treatment analysis available for a NSCLC cohort (as shown in Figure 1A). This would support the generalizability of these findings from melanoma to NSCLC. Also, please provide more information about the cohort used in the methods section. Figures 5C should include number of patients for each category (increased vs decreased) as well as median OS. As a minor suggestion, would recommend weeks be changed to months on the X axis.

- As outlined above, would be helpful to see response to triplet therapy in 393P tumors as from the reports here these also appear to develop resistance to MEK+ PD-1 combination therapy

- For triplet therapy, further control experiments would be helpful. What is the effect of IL-17 + PD-1 inhibition without MEK inhibition? Activity of IL-17 is not shown either.

Reviewer: Kathryn C. Arbour

Reviewer #2:

Remarks to the Author:

In this manuscript, Peng and colleagues demonstrate that the combination of MEK inhibition and PD-1 pathway blockade leads to transient tumor regressions in KRAS-mutant lung cancer models, but that tumors eventually develop resistance due to increased infiltration of Th17 cells. Triple therapy targeting MEK, PD-1 and IL-17 produced significant in vivo responses.

1. While MEK inhibitor plus PD-1 inhibitor combinations have shown preclinical activity in this and other manuscripts, the available clinical data has shown no added clinical activity (e.g., Hellmann, et al. Ann Oncol 2019) of the combination and many such studies have been stopped. Can the authors provide more background beyond the description of their ongoing trial?

2. Figure 2D: In the 393P KP model, AZD6244 delayed tumor growth compared to control, yet had more lung metastases. Can the authors explain this apparent disconnect?
3. Page 8: The authors repeatedly refer to Figure 2 at the top of this page, but it appears that they are referencing Figure 3?
4. The authors note that no publicly available datasets are available of lung cancer patients treated with PD-1 inhibitors with paired pre and on-treatment biopsies. As a result, they present findings from a melanoma dataset. Overall, this feels rather forced since: 1) it's a different disease, 2) patients were treatment with PD-1 monotherapy, and 3) we aren't provided with a breakdown of mutational profile of the melanoma patients.
5. Have the investigators evaluated baseline Th17-associated genes in baseline specimens from lung cancer patients treatment with PD-1 inhibitors to evaluate the relationship with response to checkpoint blockade?

We are very appreciative of the Reviewers for their support and assistance in improving our manuscript for publication and we have addressed each concern in a point-by-point manner below.

REVIEWER COMMENTS

Reviewer #1 (Remarks to the Author):

The authors here correctly point out two difficult facts in the treatment of NSCLC: 1) single agent MEK inhibition is for the most part not effective in the treatment of KRAS mutant NSCLC and 2) while immune checkpoint inhibitors have been a great development in the field, most patients will not respond to therapy. Improving response to immune checkpoint inhibitors is an important effort in the field and one certainly worthy of exploration as a topic.

Using established mouse KP model for NSCLC (KRAS mutant/TP53 mutant), that treatment with single agent MEK inhibitor leads to increased PD-L1 expression and treatment with a PD-(L)1 antibody leads to increase in MAPK signaling. Therefore, this is a potentially attractive target for combination therapy.

Shown here is that combination therapy with MEK and PD-(L) inhibition is synergistic initially but only briefly. These experiments are important ones, and clinical trials to date combining PD-L1 and MEK inhibition have not been exciting in terms of results, although many trials have been conducted in an unselected patient population.

The authors went on to evaluate potential mechanisms of resistance examining the immune infiltrate in these tumors. In the setting of resistance, the authors observed and upregulation of Th17 CD4+ T-cells are identified which secrete IL-17 and IL-22

IL-17 + MEK + PD-L1 inhibition is synergistic. This is demonstrated only in the 334SQ tumors. No experiments are shown in the 393P tumors which would be helpful to know if this combination is effective in multiple models or particular to the 334SQ tumor model. The finds the authors describe are potentially clinically interesting, given that IL-17A antibodies such as the ones used in this manuscript, are already FDA approved and could theoretically be used in a clinical trial.

We appreciate the Reviewer's comments and agree as to the limited interpretation of our initial findings since we previously evaluated the anti-IL17+AZD6244+anti-PDL1 therapy regimen in the 344SQ model. We have repeated the triple combination treatment regimen in the 393P-based model that has become resistant to MEK inhibitor treatment, shown in the revised Figure 6D. We have also now included all the treatment arms and interestingly, we found that only the triple combination is effective in controlling the tumor growth in both of these aggressively growing models.

Comments to authors:

- Is the combination shown here specifically only to the KP model (KRAS/PD53 co-mutation) or do you anticipate to be generalizable to other common co-mutation patterns which may have different responses to immunotherapy (KRAS/LKB1 mutant for example).

We have treated LKB1 mutant cell lines (A549 and H157) with the MEK inhibitor and observed an increase in PDL1 levels (supplementary Figure 1A). The basal levels of PD-L1 expression

are much lower in these cells, as we have observed previously. We do see a statistically significant increase in PD-L1, although not to the same degree as LKB1 wild type cells. Per the published literature, the mechanism of resistance in Kras/LKB1 (KL) tumors is different than in Kras/P53 (KP) tumors. KL tumors have been shown to express lower levels of immune markers, including PD-L1 (immunologically “cold” tumors), whereas KP tumors have higher levels of immune checkpoint effector molecules¹. Co-occurring genomic alterations have distinct biology and response to immune check point inhibitors. Skoulidis *et al* had found that the STK11/LKB1 alterations promote resistance to anti-PD-1 in the context of KRAS-mutant lung adenocarcinoma due to low PD-L1 expression as compared to KP tumors². Interestingly, our study also showed that 393P tumors have high basal levels of Th17 cells, potentially contributing to baseline immune evasion promoting tumor initiation and growth. This mechanism and the combination of MEK inhibitor with immune checkpoint blockade and anti-IL17 could potentially be extended in KRAS;LKB1 types and future studies utilizing syngeneic KL tumors would be valuable to test this treatment strategy.

- You report that in patients with melanoma increase in IL-17 while on treatment was associated with shorter OS. It is unfortunate that this analysis was performed in melanoma patients and not NSCLC patients. Although it is understandable that there is not a comparison cohort with NSCLC patients with pre and on-treatment tissue analysis available, is baseline pre-treatment analysis available for a NSCLC cohort (as shown in Figure 5A). This would support the generalizability of these findings from melanoma to NSCLC.

We appreciate the suggestions.

At the time we submitted our manuscript we performed an in-depth search of GEO datasets using NSCLC, homo-sapiens, and immunotherapy as keywords. We were unable to find any datasets of NSCLC immunotherapy treated patients with pre- and post-treatment biopsies. However, our group has observed that the B16 melanoma model has similar growth kinetics with immunotherapy treatment as compared to our KP models³. We therefore sought to investigate the Th17 gene signature with immunotherapy treated melanoma patient datasets (Figure 5).

A dataset (GSE126044) in NSCLC was recently made available in September 2020⁴. NSCLC patients were treated with nivolumab or pembrolizumab and biopsies were obtained. RNA-sequencing was performed on tumor samples from 4 responders or 11 non-responders. We have analyzed this new dataset for Th17 markers and have found similar trends as with the melanoma data, despite the limited numbers. This new data has been incorporated into the revised version of Figure 5A.

Also, please provide more information about the cohort used in the methods section. Figures 5C should include number of patients for each category (increased vs decreased) as well as median OS. As a minor suggestion, would recommend weeks be changed to months on the X axis.

To clarify the details about the patient tumor cohorts, we have added the number of patients for each category as well as median OS in the methods section for each respective graph. We have also changed the weeks to months on the x-axis for Figure 5D, as requested by the Reviewer.

- As outlined above, would be helpful to see response to triplet therapy in 393P tumors as from the reports here these also appear to develop resistance to MEK+ PD-1 combination therapy

We appreciate this suggestion from the Reviewer and have repeated the triple combination treatment regimen in the 393P model. This new data is incorporated into the revised version of Figure 6D. Interestingly we found that the triple combination is also dramatically effective in controlling 393P tumor growth compared to any of the single agent or double combination treatments.

- For triplet therapy, further control experiments would be helpful. What is the effect of IL-17 + PD-1 inhibition without MEK inhibition? Activity of IL-17 is not shown either.

We appreciate the Reviewer's suggestions to improve the interpretation of the *in vivo* experiments. We have repeated the experiment using each of the single and double combination therapy arms. This data is incorporated into the Revised Figure 6A and shows that neither single agent aIL17 or combination aIL17+aPDL1 control appeared to offer therapeutic benefit on par with the triple combination in the 344SQ model. We have also repeated the triple combination treatment regimen and each of the control arms in the 393P model presented in the revised Figure 6D. Again, the results highlight that only the triple combination has therapeutic efficacy against the 393P-based model.

Reviewer #2 (Remarks to the Author):

In this manuscript, Peng and colleagues demonstrate that the combination of MEK inhibition and PD-1 pathway blockade leads to transient tumor regressions in KRAS-mutant lung cancer models, but that tumors eventually develop resistance due to increased infiltration of Th17 cells. Triple therapy targeting MEK, PD-1 and IL-17 produced significant *in vivo* responses.

1. While MEK inhibitor plus PD-1 inhibitor combinations have shown preclinical activity in this and other manuscripts, the available clinical data has shown no added clinical activity (e.g., Hellmann, et al. Ann Oncol 2019) of the combination and many such studies have been stopped. Can the authors provide more background beyond the description of their ongoing trial?

We agree with the Reviewer that this dual combination therapy (MEKi plus PD-(L)1 inhibitor) has been disappointing in multiple studies across different tumor types and motivates the work presented here to understand the resistance mechanisms that limit the clinical efficacy. In the revised manuscript we have provided a more complete background of the studies and findings from the clinical trials in the Introduction section.

2. Figure 2D: In the 393P KP model, AZD6244 delayed tumor growth compared to control, yet had more lung metastases. Can the authors explain this apparent disconnect?

The 393P KP model is non-metastatic without treatment, however with extended treatment with MEK inhibitor (AZD6244) the slowly growing tumors become resistant through an epithelial-to-mesenchymal transition (EMT)⁵. The tumor cell EMT drives metastases as demonstrated by an

increase in number of lung metastatic nodules. We found that addition of aPDL1 to AZD6244 did not have an impact on the primary tumor size, but it did prevent formation of lung metastatic nodules. Based on our data we believe the combination is effective in preventing metastatic outgrowth in MEK inhibitor treatment resistant tumors in the 393P model.

3. Page 8: The authors repeatedly refer to Figure 2 at the top of this page, but it appears that they are referencing Figure 3?

We appreciate the Reviewer pointing out this discrepancy, which has been corrected in the revision.

4. The authors note that no publicly available datasets are available of lung cancer patients treated with PD-1 inhibitors with paired pre and on-treatment biopsies. As a result, they present findings from a melanoma dataset. Overall, this feels rather forced since: 1) it's a different disease, 2) patients were treatment with PD-1 monotherapy, and 3) we aren't provided with a breakdown of mutational profile of the melanoma patients.

We appreciate the reviewer's comments and have addressed each point below.

1) At the time we submitted our manuscript we performed an in-depth search of GEO datasets using NSCLC, homo-sapiens, and immunotherapy as keywords. We were unable to find any RNA sequencing datasets of NSCLC immunotherapy treated patients with pre- and post-treatment biopsies. However, our group has observed that the B16 melanoma model has similar growth kinetics with immunotherapy treatment as compared to our KP models³. We therefore sought to investigate the Th17 gene signature with immunotherapy treated melanoma patient datasets (Figure 5).

A dataset (GSE126044) in NSCLC was recently made available in September 2020⁴. NSCLC patients were treated with nivolumab or pembrolizumab and pre-treatment biopsies were obtained. RNA-sequencing was performed on tumor samples from 4 responders or 11 non-responders. We have analyzed the data for Th17 markers and have found similar trends as with the melanoma data (Figure 5A). Although there are limited number of responders, we have found this dataset promising.

2) We have observed in our KP GEMM's (Figure 3E) and syngeneic tumors (Figure 3C-D) that anti-PDL1 monotherapy has an increase in Th17 cell infiltration. Combination of MEK inhibitor to anti-PDL1 does lead to a drastic Th17 infiltration once resistance develops. We were unable to find IO and MEK inhibitor treated patient datasets for analyses, but believe that the PD1 monotherapy datasets are the best relevant and available for comparison.

3) We have evaluated the mutational profile of the samples from the melanoma dataset and found that there were no trends in terms of IL17RA, TGFB1, or CCR5 expression between the pre-treatment or on-treatment groups (data not shown) based upon the tumor oncogenotype. We have included this additional mutational profile breakdown in the revised version in supplementary Figure 6C.

5. Have the investigators evaluated baseline Th17-associated genes in baseline specimens

from lung cancer patient's treatment with PD-1 inhibitors to evaluate the relationship with response to checkpoint blockade?

We appreciate the Reviewer's suggestion. We performed an analysis of pre-treatment biopsies from NSCLC patients. We did not find any differences in Th17 gene expression. We also did not observe differences in Th17 cells at baseline in our mouse models, as shown in Figure 3C-D. Based on NSCLC dataset GSE126044 we found a difference in IL17RC expression between responders and non-responders to nivolumab Figure 5A.

Our data suggest that treatment resistance is necessary for Th17 cell infiltration. We also evaluated LUAD samples in TCGA for Th17 gene and found no consistent trends between different Kras mutations.

References

1. Skoulidis F, *et al.* Co-occurring genomic alterations define major subsets of KRAS-mutant lung adenocarcinoma with distinct biology, immune profiles, and therapeutic vulnerabilities. *Cancer Discov* **5**, 860-877 (2015).
2. Skoulidis F, *et al.* STK11/LKB1 Mutations and PD-1 Inhibitor Resistance in KRAS-Mutant Lung Adenocarcinoma. *Cancer Discov* **8**, 822-835 (2018).
3. Chen L, *et al.* CD38-Mediated Immunosuppression as a Mechanism of Tumor Cell Escape from PD-1/PD-L1 Blockade. *Cancer Discov* **8**, 1156-1175 (2018).
4. Cho J-W, *et al.* Genome-wide identification of differentially methylated promoters and enhancers associated with response to anti-PD-1 therapy in non-small cell lung cancer. *Experimental & Molecular Medicine* **52**, 1550-1563 (2020).
5. Peng DH, *et al.* ZEB1 suppression sensitizes KRAS mutant cancers to MEK inhibition by an IL17RD-dependent mechanism. *Sci Transl Med* **11**, (2019).

Reviewers' Comments:

Reviewer #1:

Remarks to the Author:

The authors have adequately and thoughtfully addressed the concerns I raised in the initial review, both in the text as well as with additional experiments.

Reviewer #2:

Remarks to the Author:

I applaud the authors for making a number of changes in the manuscript based upon reviewer feedback. Based upon the changes, I would add the following:

- 1). Several LKB1 mutant cell lines (H157 and A549) were added to the manuscript but the results section doesn't specify that these are LKB1 mutant, it simply states "lung cancer cell lines harboring varying co-mutations." It's therefore confusing to the reader when we get to the discussion (lines 338-345), which begins "The observation that KL cell lines increase PD-L1 levels is significant..." This was effectively the first mention of STK11 mutations.
- 2). The authors have greatly improved the manuscript by adding a NSCLC clinical cohort, but the total number of samples should be listed in the text as this is a very small dataset.
- 3). The penultimate sentence of the abstract should be modified to add the NSCLC data (and make clear that the pre- and post- were melanoma samples).
- 4). Minor point - Figure 5A uses labels of non-responder versus responder while 5B uses labels of PD/SD versus PR/CR. These should be consistent.

We are very thankful and appreciative of the reviewers for their support and assistance in improving our manuscript for publication and we have addressed each concern in a point-by-point manner below.

Reviewer #1 (Remarks to the Author):

The authors have adequately and thoughtfully addressed the concerns I raised in the initial review, both in the text as well as with additional experiments.

We would like to thank the reviewer for their suggestions and giving us the opportunity to improve our manuscript significantly.

Reviewer #2 (Remarks to the Author):

I applaud the authors for making a number of changes in the manuscript based upon reviewer feedback. Based upon the changes, I would add the following:

We would like to thank the reviewer for their suggestions and giving us the opportunity to improve our manuscript significantly.

1). Several LKB1 mutant cell lines (H157 and A549) were added to the manuscript but the results section doesn't specify that these are LKB1 mutant, it simply states "lung cancer cell lines harboring varying co-mutations." It's therefore confusing to the reader when we get to the discussion (lines 338-345), which begins "The observation that KL cell lines increase PD-L1 levels is significant..." This was effectively the first mention of STK11 mutations.

We have clarified the LKB1 mutant cell line in the results section, line 112-114.

2). The authors have greatly improved the manuscript by adding a NSCLC clinical cohort, but the total number of samples should be listed in the text as this is a very small dataset.

We have added the total number of samples per group in the text, line 258-259.

3). The penultimate sentence of the abstract should be modified to add the NSCLC data (and make clear that the pre- and post- were melanoma samples).

We have modified the abstract to clarify the specific samples, line 37-39.

4). Minor point – Figure 5A uses labels of non-responder versus responder while 5B uses labels of PD/SD versus PR/CR. These should be consistent.

Cho et al. 2020 used RECIST criteria to define patients as responders if they showed partial response (PR) and non-responders if they showed progressive disease (PD). We can use PR and PD as that is how the authors defined the patients, we cannot use PD/SD or PR/CR as the author did not classify patients based upon the criteria set up by Riaz et al. 2017.